# PRE-TRAINING VIA DENOISING
# FOR MOLECULAR PROPERTY PREDICTION

**Sheheryar Zaidi**[*†§]  **Michael Schaarschmidt**[*‡]
**James Martens**[‡]  **Hyunjik Kim**[‡]  **Yee Whye Teh**[‡]  **Alvaro Sanchez-Gonzalez**[‡]
**Peter Battaglia**[‡]  **Razvan Pascanu**[‡]  **Jonathan Godwin**[‡]
[†]University of Oxford, [‡]DeepMind
[*]Equal contribution. [§]Work done during an internship at DeepMind.
Correspondence to: `shehzaidi@deepmind.com`, `mschaarschmidt@google.com`.

## ABSTRACT

Many important problems involving molecular property prediction from 3D structures have limited data, posing a generalization challenge for neural networks. In this paper, we describe a pre-training technique based on denoising that achieves a new state-of-the-art in molecular property prediction by utilizing large datasets of 3D molecular structures at equilibrium to learn meaningful representations for downstream tasks. Relying on the well-known link between denoising autoencoders and score-matching, we show that the denoising objective corresponds to learning a molecular force field – arising from approximating the Boltzmann distribution with a mixture of Gaussians – directly from equilibrium structures. Our experiments demonstrate that using this pre-training objective significantly improves performance on multiple benchmarks, achieving a new state-of-the-art on the majority of targets in the widely used QM9 dataset. Our analysis then provides practical insights into the effects of different factors – dataset sizes, model size and architecture, and the choice of upstream and downstream datasets – on pre-training.

## 1 INTRODUCTION

The success of the best performing neural networks in vision and natural language processing (NLP) relies on pre-training the models on large datasets to learn meaningful features for downstream tasks (Dai & Le, 2015; Simonyan & Zisserman, 2014; Devlin et al., 2018; Brown et al., 2020; Dosovitskiy et al., 2020). For molecular property prediction from 3D structures (a point cloud of atomic nuclei in $\mathbb{R}^3$), the problem of how to similarly learn such representations remains open. For example, none of the best models on the widely used QM9 benchmark use any form of pre-training (*e.g.* Klicpera et al., 2020a; Liu et al., 2022b; Schütt et al., 2021; Thölke & De Fabritiis, 2022), in stark contrast with vision and NLP. Effective methods for pre-training could have a significant impact on fields such as drug discovery and material science.

In this work, we focus on the problem of how large datasets of 3D molecular structures can be utilized to improve performance on downstream molecular property prediction tasks that also rely on 3D structures as input. We address the question: how can one exploit large datasets like PCQM4Mv2,[1] that contain over 3 million structures, to improve performance on datasets such as DES15K that are orders of magnitude smaller? Our answer is a form of self-supervised pre-training that generates useful representations for downstream prediction tasks, leading to state-of-the-art (SOTA) results.

Inspired by recent advances in noise regularization for graph neural networks (GNNs) (Godwin et al., 2022), our pre-training objective is based on denoising in the space of structures (and is hence self-supervised). Unlike existing pre-training methods, which largely focus on 2D graphs, our approach targets the setting where the downstream task involves 3D point clouds defining the molecular structure. Relying on the well-known connection between denoising and score-matching (Vincent, 2011; Song & Ermon, 2019; Ho et al., 2020), we show that the denoising objective is equivalent to learning a particular force field, adding a new interpretation of denoising in the context of molecules and shedding light on how it aids representation learning.

---

[1]Note that PCQM4Mv2 is a new version of PCQM4M that now offers 3D structures.

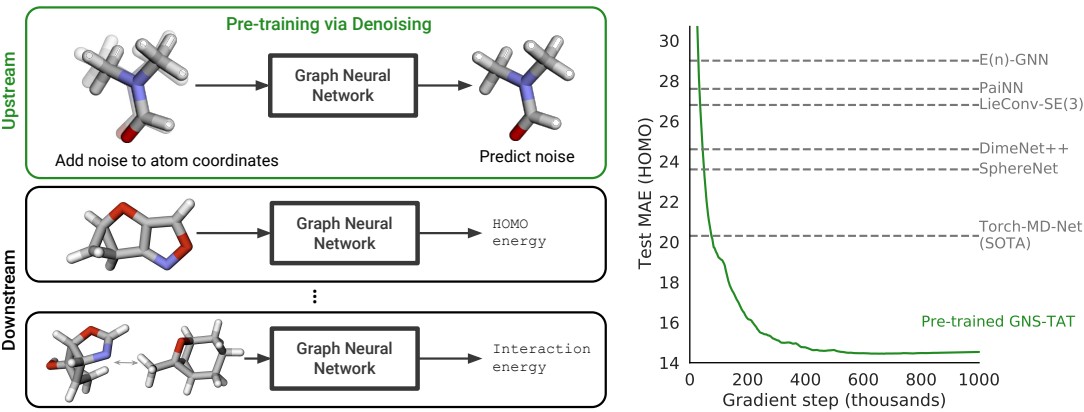

Figure 1: GNS-TAT pre-trained via denoising on PCQM4Mv2 outperforms prior work on QM9.

The contributions of our work are summarized as follows:

- We investigate a simple and effective method for pre-training via denoising in the space of 3D structures with the aim of improving downstream molecular property prediction from such 3D structures. Our denoising objective is shown to be related to learning a specific force field.

- Our experiments demonstrate that pre-training via denoising significantly improves performance on multiple challenging datasets that vary in size, nature of task, and molecular composition. This establishes that denoising over structures successfully transfers to molecular property prediction, setting, in particular, a new state-of-the-art on 10 out of 12 targets in the widely used QM9 dataset. Figure 1 illustrates performance on one of the targets in QM9.

- We make improvements to a common GNN, in particular showing how to apply Tailored Activation Transformation (TAT) (Zhang et al., 2022) to Graph Network Simulators (GNS) (Sanchez-Gonzalez et al., 2020), which is complementary to pre-training and further boosts performance.

- We analyze the benefits of pre-training by gaining insights into the effects of dataset size, model size and architecture, and the relationship between the upstream and downstream datasets.

## 2 RELATED WORK

**Pre-training of GNNs.** Various recent works have formulated methods for pre-training using graph data (Liu et al., 2021b; Hu et al., 2020a; Xie et al., 2021; Kipf & Welling, 2016), rather than 3D point clouds of atom nuclei as in this paper. Approaches based on contrastive methods rely on learning representations by contrasting different *views* of the input graph (Sun et al., 2019; Veličković et al., 2019; You et al., 2020; Liu et al., 2021a), or bootstrapping (Thakoor et al., 2021). Autoregressive or reconstruction-based approaches, such as ours, learn representations by requiring the model to predict aspects of the input graph (Hu et al., 2020a;b; Rong et al., 2020; Liu et al., 2019). Most methods in the current literature are not designed to handle 3D structural information, focusing instead on 2D graphs. The closest work to ours is GraphMVP (Liu et al., 2021a), where 3D structure is treated as one view of a 2D molecule for the purpose of upstream contrastive learning. Their work focuses on downstream tasks that only involve 2D information, while our aim is to improve downstream models for molecular property prediction from 3D structures. After the release of this pre-print, similar ideas have been studied by Jiao et al. (2022) and Liu et al. (2022a).

**Denoising, representation learning and score-matching.** Noise has long been known to improve generalization in machine learning (Sietsma & Dow, 1991; Bishop, 1995). Denoising autoencoders have been used to effectively learn representations by mapping corrupted inputs to original inputs (Vincent et al., 2008; 2010). Specific to GNNs (Battaglia et al., 2018; Scarselli et al., 2009; Bronstein et al., 2017), randomizing input graph features has been shown to improve performance (Hu et al., 2020a; Sato et al., 2021). Applications to physical simulation also involve corrupting the state with Gaussian noise (Sanchez-Gonzalez et al., 2018; 2020; Pfaff et al., 2020). Our work builds on Noisy Nodes (Godwin et al., 2022), which incorporates denoising as an auxiliary task to improve performance, indicating the effectiveness of denoising for molecular property prediction (*cf.* Section 3.2.2). Denoising is also closely connected to score-matching (Vincent, 2011), which has become popular

for generative modelling (Song & Ermon, 2019; 2020; Ho et al., 2020; Hoogeboom et al., 2022; Xu et al., 2022; Shi et al., 2021). We also rely on this connection to show that denoising structures corresponds to learning a force field.

**Equivariant neural networks for 3D molecular property prediction.** Recently, the dominant approach for improving models for molecular property prediction from 3D structures has been through the design of architectures that incorporate roto-translational inductive biases into the model, such that the outputs are invariant to translating and rotating the input atomic positions. A simple way to achieve this is to use roto-translation invariant features as inputs, such as inter-atomic distances (Schütt et al., 2017; Unke & Meuwly, 2019), angles (Klicpera et al., 2020b;a; Shuaibi et al., 2021; Liu et al., 2022b), or the principal axes of inertia (Godwin et al., 2022). There is also broad literature on equivariant neural networks, whose intermediate activations transform accordingly with roto-translations of inputs thereby naturally preserving inter-atomic distance and orientation information. Such models can be broadly categorized into those that are specifically designed for molecular property prediction (Thölke & De Fabritiis, 2022; Schütt et al., 2021; Batzner et al., 2021; Anderson et al., 2019; Miller et al., 2020) and general-purpose architectures (Satorras et al., 2021; Finzi et al., 2020; Hutchinson et al., 2021; Thomas et al., 2018; Kondor et al., 2018; Brandstetter et al., 2021). Our pre-training technique is architecture-agnostic, and we show that it can be applied to enhance performance in both a GNN-based architecture (Sanchez-Gonzalez et al., 2020) and a Transformer-based one (Thölke & De Fabritiis, 2022). We conjecture that similar improvements will hold for other models.

## 3 METHODOLOGY

### 3.1 PROBLEM SETUP

Molecular property prediction consists of predicting scalar quantities given the structure of one or more molecules as input. Each data example is a labelled set specified as follows: we are provided with a set of atoms $S = \{(a_1, \mathbf{p}_1), \ldots, (a_{|S|}, \mathbf{p}_{|S|})\}$, where $a_i \in \{1, \ldots, 118\}$ and $\mathbf{p}_i \in \mathbb{R}^3$ are the atomic number and 3D position respectively of atom $i$ in the molecule, alongside a label $y \in \mathbb{R}$. We assume that the model, which takes $S$ as input, is any architecture consisting of a backbone, which first processes $S$ to build a latent representation of it, followed by a vertex-level or graph-level "decoder", that returns per-vertex predictions or a single prediction for the input respectively.

### 3.2 PRE-TRAINING VIA DENOISING

Given a dataset of molecular structures, we pre-train the network by denoising the structures, which operates as follows. Let $\mathcal{D}_{\text{structures}} = \{S_1, \ldots, S_n\}$ denote the upstream dataset of equilibrium structures, and let $\text{GNN}_\theta$ denote a graph neural network with parameters $\theta$ which takes $S \in \mathcal{D}_{\text{structures}}$ as input and returns per-vertex predictions $\text{GNN}_\theta(S) = (\hat{\boldsymbol{\epsilon}}_1, \ldots, \hat{\boldsymbol{\epsilon}}_{|S|})$. The precise parameterization of the models we consider in this work is described in Section 3.3 and Appendix A.

Starting with an input molecule $S \in \mathcal{D}_{\text{structures}}$, we perturb it by adding i.i.d. Gaussian noise to its atomic positions $\mathbf{p}_i$. That is, we create a noisy version of the molecule:

$$\tilde{S} = \{(a_1, \tilde{\mathbf{p}}_1), \ldots, (a_{|S|}, \tilde{\mathbf{p}}_{|S|})\}, \text{ where } \tilde{\mathbf{p}}_i = \mathbf{p}_i + \sigma \boldsymbol{\epsilon}_i \text{ and } \boldsymbol{\epsilon}_i \sim \mathcal{N}(0, I_3), \tag{1}$$

The noise scale $\sigma$ is a tuneable hyperparameter (an interpretation of which is given in Section 3.2.1). We train the model as a denoising autoencoder by minimizing the following loss with respect to $\theta$:

$$\mathbb{E}_{p(\tilde{S},S)} \left[ \left\| \text{GNN}_\theta(\tilde{S}) - (\boldsymbol{\epsilon}_1, \ldots, \boldsymbol{\epsilon}_{|S|}) \right\|^2 \right]. \tag{2}$$

The distribution $p(\tilde{S}, S)$ corresponds to sampling a structure $S$ from $\mathcal{D}_{\text{structures}}$ and adding noise to it according to Equation (1). Note that the model predicts the noise, not the original coordinates. Next, we motivate denoising as our pre-training objective for molecular modelling.

### 3.2.1 DENOISING AS LEARNING A FORCE FIELD

Datasets in quantum chemistry are typically generated by minimizing expensive-to-compute inter-atomic forces with methods such as density functional theory (DFT) (Parr & Weitao, 1994). We speculate that learning this *force field* would give rise to useful representations for downstream tasks, since molecular properties vary with forces and energy. Therefore, a reasonable pre-training objective

would be one that involves learning the force field. Unfortunately, this force field is either unknown or expensive to evaluate, and hence it cannot be used directly for pre-training. An alternative is to approximate the data-generating force field with one that can be cheaply evaluated and use it to learn good representations – an approach we outline in this section. Using the well-known link between denoising autoencoders and score-matching (Vincent, 2011; Song & Ermon, 2019; 2020), we can show that the denoising objective in Equation (2) is equivalent to learning a particular force field directly from equilibrium structures with some desirable properties. For clarity, in this subsection we condition on and suppress the atom types and molecule size in our notation, specifying a molecular structure by its coordinates $\mathbf{x} \in \mathbb{R}^{3N}$ (with $N$ as the size of the molecule).

From the perspective of statistical physics, a structure $\mathbf{x}$ can be treated as a random quantity sampled from the Boltzmann distribution $p_{\text{physical}}(\mathbf{x}) \propto \exp(-E(\mathbf{x}))$, where $E(\mathbf{x})$ is the (potential) energy of $\mathbf{x}$. According to $p_{\text{physical}}$, low energy structures have a high probability of occurring. Moreover, the per-atom forces are given by $\nabla_{\mathbf{x}} \log p_{\text{physical}}(\mathbf{x}) = -\nabla_{\mathbf{x}} E(\mathbf{x})$, which is referred to as the force field. Our goal is to learn this force field. However both the energy function $E$ and distribution $p_{\text{physical}}$ are unknown, and we only have access to a set of equilibrium structures $\mathbf{x}_1, \ldots, \mathbf{x}_n$ that locally minimize the energy $E$. Since $\mathbf{x}_1, \ldots, \mathbf{x}_n$ are then local maxima of the distribution $p_{\text{physical}}$, our main approximation is to replace $p_{\text{physical}}$ with a mixture of Gaussians centered at the data:

$$p_{\text{physical}}(\tilde{\mathbf{x}}) \approx q_\sigma(\tilde{\mathbf{x}}) := \frac{1}{n} \sum_{i=1}^{n} q_\sigma(\tilde{\mathbf{x}} \mid \mathbf{x}_i),$$

where we define $q_\sigma(\tilde{\mathbf{x}} \mid \mathbf{x}_i) = \mathcal{N}(\tilde{\mathbf{x}}; \mathbf{x}_i, \sigma^2 I_{3N})$. This approximation captures the fact that $p_{\text{physical}}$ will have local maxima at the equilibrium structures, vary smoothly with $\mathbf{x}$ and is computationally convenient. Learning the force field corresponding to $q_\sigma(\tilde{\mathbf{x}})$ now yields a score-matching objective:

$$\mathbb{E}_{q_\sigma(\tilde{\mathbf{x}})} \left[ \|\text{GNN}_\theta(\tilde{\mathbf{x}}) - \nabla_{\tilde{\mathbf{x}}} \log q_\sigma(\tilde{\mathbf{x}})\|^2 \right]. \tag{3}$$

As shown by Vincent (2011), and recently applied to generative modelling (Song & Ermon, 2019; 2020; Ho et al., 2020; Shi et al., 2021; Xu et al., 2022), this objective is equivalent to the denoising objective. Specifically, defining $q_0(\mathbf{x}) = \frac{1}{n} \sum_{i=1}^{n} \delta(\mathbf{x} = \mathbf{x}_i)$ to be the empirical distribution and $q_\sigma(\tilde{\mathbf{x}}, \mathbf{x}) = q_\sigma(\tilde{\mathbf{x}} \mid \mathbf{x}) q_0(\mathbf{x})$, the objective in Equation (3) is equivalent to:

$$\mathbb{E}_{q_\sigma(\tilde{\mathbf{x}}, \mathbf{x})} \left[ \|\text{GNN}_\theta(\tilde{\mathbf{x}}) - \nabla_{\tilde{\mathbf{x}}} \log q_\sigma(\tilde{\mathbf{x}} \mid \mathbf{x})\|^2 \right] = \mathbb{E}_{q_\sigma(\tilde{\mathbf{x}}, \mathbf{x})} \left[ \left\| \text{GNN}_\theta(\tilde{\mathbf{x}}) - \frac{\mathbf{x} - \tilde{\mathbf{x}}}{\sigma^2} \right\|^2 \right]. \tag{4}$$

We notice that the RHS corresponds to the earlier denoising loss in Equation (2) (up to a constant factor of $1/\sigma$ applied to $\text{GNN}_\theta$ that can be absorbed into the network). To summarize, denoising equilibrium structures corresponds to learning the force field that arises from approximating the distribution $p_{\text{physical}}$ with a mixture of Gaussians. Note that we can interpret the noise scale $\sigma$ as being related to the sharpness of $p_{\text{physical}}$ or $E$ around the local maxima $\mathbf{x}_i$. We also remark that the equivalence between Equation (3) and the LHS of Equation (4) does not require $q_\sigma(\tilde{\mathbf{x}} \mid \mathbf{x}_i)$ to be a Gaussian distribution (Vincent, 2011), and other choices will lead to different denoising objectives, which we leave as future work. See Appendix B for technical caveats and a discussion of the differences between denoising for generative modelling versus learning forces.

### 3.2.2 NOISY NODES: DENOISING AS AN AUXILIARY LOSS

Recently, Godwin et al. (2022) also applied denoising as an *auxiliary* loss to molecular property prediction, achieving significant improvements on a variety of molecular datasets. In particular, their approach, called Noisy Nodes, consisted of augmenting the usual optimization objective for predicting $y$ with an auxiliary denoising loss. They suggested two explanations for why Noisy Nodes improves performance. First, the presence of a vertex-level loss discourages *oversmoothing* (Chen et al., 2019; Cai & Wang, 2020) of vertex/edge features after multiple message-passing layers – a common problem plaguing GNNs – because successful denoising requires diversity amongst vertex features in order to match the diversity in the noise targets $\epsilon_i$. Second, they argued that denoising can aid representation learning by encouraging the network to learn aspects of the input distribution.

The empirical success of Noisy Nodes indicates that denoising can indeed result in meaningful representations. Since Noisy Nodes incorporates denoising only as an auxiliary task, the representation learning benefits of denoising are limited to the downstream dataset on which it is used as an auxiliary task. Our approach is to apply denoising as a pre-training objective on another large (unlabelled) dataset of structures to learn higher-quality representations, which results in better performance.

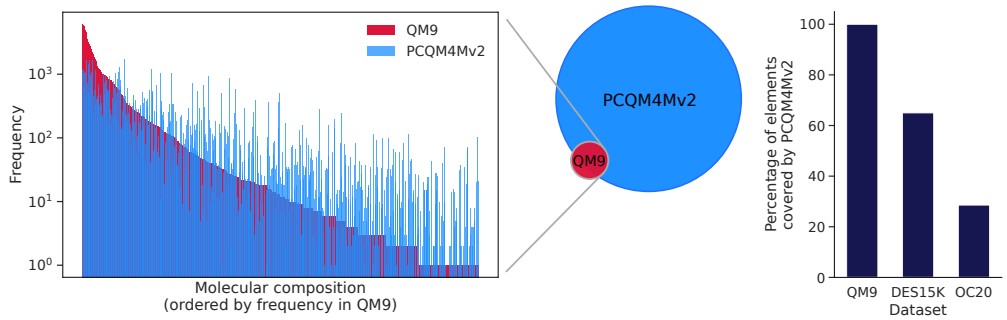

Figure 2: **Left:** Frequency of compositions of molecules appearing in QM9 overlayed with the corresponding frequency in PCQM4Mv2. Each bar represents one molecular composition (*e.g.* one carbon atom, two oxygen atoms). **Right:** Percentage of elements appearing in QM9, DES15K, OC20 that also appear in PCQM4Mv2.

## 3.3 GNS AND GNS-TAT

The main two models we consider in this work are Graph Net Simulator (GNS) (Sanchez-Gonzalez et al., 2020), which is a type of GNN, and a better-performing variant we contribute called GNS-TAT. GNS-TAT makes use of a recently published network transformation method called Tailored Activation Transforms (TAT) (Zhang et al., 2022), which has been shown to prevent certain degenerate behaviors at initialization in deep MLPs/convnets that are reminiscent of oversmoothing in GNNs (and are also associated with training difficulties). While GNS is not by default compatible with the assumptions of TAT, we propose a novel GNN initialization scheme called "Edge-Delta" that makes it compatible by initializing to zero the weights that carry "messages" from vertices to edges. This marks the first application of TAT to any applied problem in the literature. See Appendix A for details.

## 4 EXPERIMENTS

The goal of our experimental evaluation in this section is to answer the following questions. First, does pre-training a neural network via denoising improve performance on the downstream task compared to training from a random initialization? Second, how does the benefit of pre-training depend on the relationship between the upstream and downstream datasets? Our evaluation involves four realistic and challenging molecular datasets, which vary in size, compound compositions (organic or inorganic) and labelling methodology (DFT- or CCSD(T)-generated), as described below.

### 4.1 DATASETS AND TRAINING SETUP

**Datasets.** First, the main dataset we use for pre-training is **PCQM4Mv2** (Nakata & Shimazaki, 2017), which contains 3.4 million organic molecules, specified by their 3D structures at equilibrium calculated using DFT.[2] The molecules in PCQM4Mv2 only contain one label, however the labels are not used as denoising only requires the structures. The large scale and diversity of PCQM4Mv2 makes it well-suited for pre-training via denoising. Second, as a dataset for fine-tuning, we use **QM9** (Ramakrishnan et al., 2014), which contains around 130,000 small organic molecules and is widely used as a molecular property prediction benchmark (Klicpera et al., 2020a; Fuchs et al., 2020; Satorras et al., 2021; Finzi et al., 2020; Hutchinson et al., 2021; Schütt et al., 2021; Thölke & De Fabritiis, 2022; Godwin et al., 2022). Each molecule is specified by its structure alongside 12 associated molecular property labels. Third, **Open Catalyst 2020 (OC20)** (Chanussot* et al., 2021) is a recent large benchmark of interacting surfaces and adsorbates relevant to catalyst discovery. OC20 contains various tasks, such as predicting the relaxed state energy from an initial high-energy structure (IS2RE). We explore different combinations of upstream and downstream tasks as described in Section 4.3. Lastly, **DES15K** (Donchev et al., 2021) is a small dataset we use for fine-tuning, which contains around 15,000 dimer geometries (*i.e.* molecule pairs) with non-covalent molecular

---

[2]An earlier version of this dataset without any 3D structures, called PCQM4M, was used for supervised pre-training (Ying et al., 2021), but to our knowledge, this is the first time the 3D structures from v2 have been used and in a self-supervised manner.

Table 1: Results on QM9 comparing the performance of GNS-TAT + Noisy Nodes (NN) with and without pre-training on PCQM4Mv2 (averaged over three seeds) with other baselines.

| Target | Unit | SchNet | E(n)-GNN | DimeNet++ | SphereNet | PaiNN | TorchMD-NET | GNS + NN | GNS-TAT + NN | Pre-trained GNS-TAT + NN |
|---|---|---|---|---|---|---|---|---|---|---|
| $\mu$ | D | 0.033 | 0.029 | 0.030 | 0.027 | 0.012 | **0.011** | 0.025 | 0.021 | 0.016 |
| $\alpha$ | $a_0^3$ | 0.235 | 0.071 | 0.043 | 0.047 | 0.045 | 0.059 | 0.052 | 0.047 | **0.040** |
| $\epsilon_{HOMO}$ | meV | 41.0 | 29.0 | 24.6 | 23.6 | 27.6 | 20.3 | 20.4 | 17.3 | **14.9** |
| $\epsilon_{LUMO}$ | meV | 34.0 | 25.0 | 19.5 | 18.9 | 20.4 | 18.6 | 17.5 | 17.1 | **14.7** |
| $\Delta\epsilon$ | meV | 63.0 | 48.0 | 32.6 | 32.3 | 45.7 | 36.1 | 28.6 | 25.7 | **22.0** |
| $\langle R^2 \rangle$ | $a_0^2$ | 0.07 | 0.11 | 0.33 | 0.29 | 0.07 | **0.033** | 0.70 | 0.65 | 0.44 |
| ZPVE | meV | 1.700 | 1.550 | 1.210 | 1.120 | 1.280 | 1.840 | 1.160 | 1.080 | **1.018** |
| $U_0$ | meV | 14.00 | 11.00 | 6.32 | 6.26 | 5.85 | 6.15 | 7.30 | 6.39 | **5.76** |
| $U$ | meV | 19.00 | 12.00 | 6.28 | 7.33 | 5.83 | 6.38 | 7.57 | 6.39 | **5.76** |
| $H$ | meV | 14.00 | 12.00 | 6.53 | 6.40 | 5.98 | 6.16 | 7.43 | 6.42 | **5.79** |
| $G$ | meV | 14.00 | 12.00 | 7.56 | 8.00 | 7.35 | 7.62 | 8.30 | 7.41 | **6.90** |
| $c_v$ | $\frac{cal}{mol\,K}$ | 0.033 | 0.031 | 0.023 | 0.022 | 0.024 | 0.026 | 0.025 | 0.022 | **0.020** |

interactions. Each pair is labelled with its interaction energy computed using the gold-standard CCSD(T) method (Bartlett & Musiał, 2007). CCSD(T) is usually both more expensive and accurate than DFT, which is used for all aforementioned datasets. See Appendix D for further details and a discussion about the choice of using DFT-generated structures for pre-training.

Figure 2 (right) shows what percentage of elements appearing in each of QM9, OC20 and DES15K also appear in PCQM4Mv2. Whereas QM9 is fully covered by PCQM4Mv2, we observe that DES15K has less element overlap with PCQM4Mv2 and less than < 30% of elements in OC20 are contained in PCQM4Mv2. This is owing to the fact that surface molecules in OC20 are inorganic lattices, none of which appear in PCQM4Mv2. This suggests that we can expect least transfer from PCQM4Mv2 to OC20. We also compare PCQM4Mv2 and QM9 in terms of the molecular compositions, *i.e.* the number of atoms of each element, that appear in each. Due to presence of isomers, both datasets contain multiple molecules with the same composition. For each molecular composition in QM9, Figure 2 (left) shows its frequency in both QM9 and PCQM4Mv2. We observe that most molecular compositions in QM9 also appear in PCQM4Mv2. We also remark that since pre-training is self-supervised using only unlabelled structures, test set contamination is not possible – in fact, PCQM4Mv2 does not have most of the labels in QM9.

**Training setup.** GNS/GNS-TAT were implemented in JAX (Bradbury et al., 2018) using Haiku and Jraph (Hennigan et al., 2020; Godwin* et al., 2020). All experiments were averaged over 3 seeds. Detailed hyperparameter and hardware settings can be found in Appendices E and F.

## 4.2 RESULTS ON QM9

We evaluate two variants of our model on QM9 in Table 1, GNS-TAT with Noisy Nodes trained from a random initialization versus pre-trained parameters. Pre-training is done on PCQM4Mv2 via denoising. For best performance on QM9, we found that using atom type masking and prediction during pre-training additionally helped (Hu et al., 2020a). We fine-tune a separate model for each of the 12 targets, as usually done on QM9, using a single pre-trained model. This is repeated for three seeds (including pre-training). Following customary practice, hyperparameters, including the noise scale for denoising during pre-training and fine-tuning, are tuned on the HOMO target and then kept fixed for all other targets. We first observe that GNS-TAT with Noisy Nodes performs competitively with other models and significantly improves upon GNS with Noisy Nodes, revealing the benefit of the TAT modifications. Utilizing pre-training then further improves performance across all targets, achieving a new state-of-the-art compared to prior work for 10 out of 12 targets. Interestingly, for the electronic spatial extent target $\langle R^2 \rangle$, we found GNS-TAT to perform worse than other models, which may be due to the optimal noise scale being different from that of other targets.

## 4.3 RESULTS ON OC20

Next, we consider the Open Catalyst 2020 benchmark focusing on the downstream task of predicting the relaxed energy from the initial structure (IS2RE). We compared GNS with Noisy Nodes trained

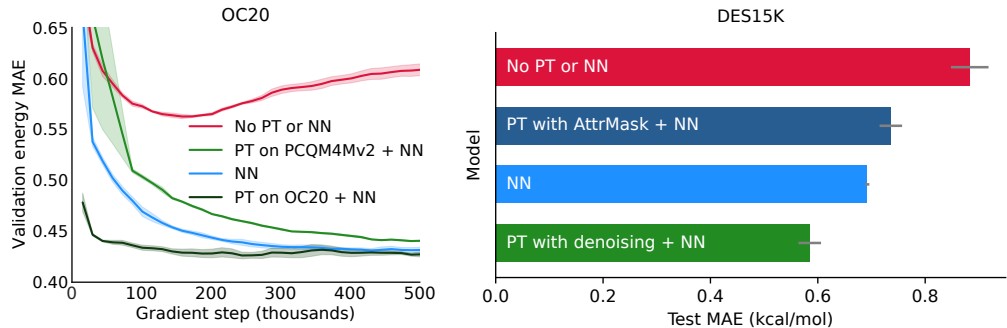

Figure 3: **Left:** Validation performance curves on the OC20 IS2RE task (`ood_both` split) with GNS. See Table 9 for a comparison to other models in the literature. **Right:** Test performance curves for predicting interaction energies of dimer geometries in the DES15K dataset with GNS-TAT. "PT" and "NN" stand for pre-training and Noisy Nodes respectively.

from scratch versus using pre-trained parameters. We experimented with two options for pre-training: (1) pre-training via denoising on PCQM4Mv2, and (2) pre-training via denoising on OC20 itself. For the latter, we follow Godwin et al.'s [2022] approach of letting the denoising target be the relaxed structure, while the perturbed input is a random interpolation between the initial and relaxed structures with added Gaussian noise – this corresponds to the IS2RS task with additional noise. As shown in Figure 3 (left), pre-training on PCQM4Mv2 offers no benefit for validation performance on IS2RE, however pre-training on OC20 leads to considerably faster convergence but the same final performance. The lack of transfer from PCQM4Mv2 to OC20 is likely due to the difference in nature of the two datasets and the small element overlap as discussed in Section 4.1 and Figure 2 (right). On the other hand, faster convergence from using parameters pre-trained on OC20 suggests that denoising learned meaningful features. Unsurprisingly, the final performance is unchanged since the upstream and downstream datasets are the same in this case, so pre-training with denoising is identical to the auxiliary task of applying Noisy Nodes. The performance achieved is also competitive with other models in the literature as shown in Table 9.

## 4.4 RESULTS ON DES15K

In our experiments so far, all downstream tasks were based on DFT-generated datasets. While DFT calculations are more expensive than using neural networks, they are relatively cheap compared to even higher quality methods such as CCSD(T) (Bartlett & Musiał, 2007). In this section, we evaluate how useful pre-training on DFT-generated structures from PCQM4Mv2 is when fine-tuning on the recent DES15K dataset which contains higher quality CCSD(T)-generated interaction energies. Moreover, unlike QM9, inputs from DES15K are systems of two interacting molecules and the dataset contains only around 15,000 examples, rendering it more challenging. We compare the test performance on DES15K achieved by GNS-TAT with Noisy Nodes when trained from scratch versus using pre-trained parameters from PCQM4Mv2. As a baseline, we also include pre-training on PCQM4Mv2 using 2D-based AttrMask (Hu et al., 2020a) by masking and predicting atomic numbers. Figure 3 (right) shows that using Noisy Nodes significantly improves performance compared to training from scratch, with a further improvement resulting from using pre-training via denoising. AttrMask underperforms denoising since it likely does not fully exploit the 3D structural information. Importantly, this shows that pre-training by denoising structures obtained through relatively cheap methods such as DFT can even be beneficial when fine-tuning on more expensive and smaller downstream datasets. See Appendix G.1 for similar results on another architecture.

## 5 ANALYSIS

### 5.1 PRE-TRAINING A DIFFERENT ARCHITECTURE

To explore whether pre-training is beneficial beyond GNS/GNS-TAT, we applied pre-training via denoising to the TorchMD-NET architecture (Thölke & De Fabritiis, 2022). TorchMD-NET is a transformer-based architecture whose layers maintain per-atom scalar features $x_i \in \mathbb{R}^F$ and vector

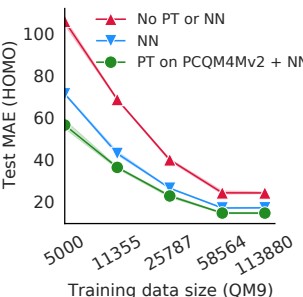 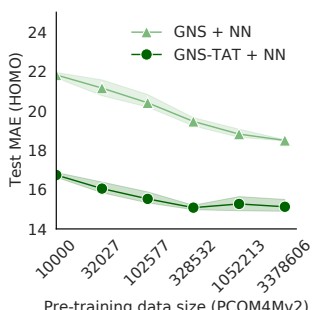 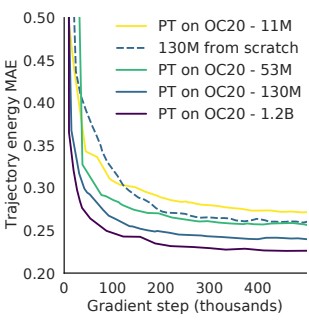

Figure 4: **Left:** Impact of varying the downstream dataset size for the HOMO target in QM9 with GNS-TAT. **Middle:** Impact of varying the upstream dataset size for the HOMO target in QM9. **Right:** Validation performance curves on the OC20 S2EF task (`ood_both` split) for different GNS model sizes. "PT" and "NN" stand for pre-training and Noisy Nodes respectively.

features $v_i \in \mathbb{R}^{3 \times F}$, where $F$ is the feature dimension, that are updated in each layer using a self-attention mechanism. We implemented denoising by using gated equivariant blocks (Weiler et al., 2018; Schütt et al., 2021) applied to the processed scalar and vector features. The resulting vector features are then used as the noise prediction.

In Table 2, we evaluate the effect of adding Noisy Nodes and pre-training to the architecture on the HOMO and LUMO targets in QM9. Pre-training yields a boost in performance, allowing the model to achieve SOTA results. Note that the results shown for TorchMD-NET are from our runs using Thölke & De Fabritiis's [2022] open-source code, which led to slightly worse results than their

Table 2: Performance of TorchMD-NET with Noisy Nodes and pre-training on PCQM4Mv2.

| Method | $\epsilon_{\text{HOMO}}$ | $\epsilon_{\text{LUMO}}$ |
|---|---|---|
| TorchMD-NET | $22.0 \pm 0.6$ | $18.7 \pm 0.4$ |
| + Noisy Nodes | $18.1 \pm 0.1$ | $15.6 \pm 0.1$ |
| + Pre-training | $\mathbf{15.6 \pm 0.1}$ | $\mathbf{13.2 \pm 0.2}$ |

published ones (our pre-training results still outperform their published results). Our code for experiments on TorchMD-NET is open source.[3]

## 5.2 VARYING DATASET SIZES

We also investigate how downstream test performance on the HOMO target in QM9 varies as a function of the number of upstream and downstream training examples. First, we compare the performance of GNS-TAT with Noisy Nodes either trained from scratch or using pre-trained parameters for different numbers of training examples from QM9; we also include the performance of just GNS-TAT. As shown in Figure 4 (left), pre-training improves the downstream performance for all dataset sizes. The difference in test MAE also grows as the downstream training data reduces. Second, we assess the effect of varying the amount of pre-training data while fixing the downstream dataset size for both GNS and GNS-TAT as shown in Figure 4 (middle). For both models, we find that downstream performance generally improves as upstream data increases, with saturating performance for GNS-TAT. More upstream data can yield better quality representations.

## 5.3 VARYING MODEL SIZE

We study the benefit of pre-training as models are scaled up on large downstream datasets. Recall that the S2EF dataset in OC20 contains around 130 million DFT evaluations for catalytic systems, providing three orders of magnitude more training data than QM9. We compare the performance of four GNS models with sizes ranging from 10 million to 1.2 billion parameters scaled up by increasing the hidden layer sizes in the MLPs. Each is pre-trained via denoising using the trajectories provided for the IS2RE/IS2RS tasks as described in Section 4.3. We also compare this to a 130 million parameter variant of GNS trained from scratch. As shown in Figure 4 (right), the pre-trained models continue to benefit from larger model sizes. We also observe that pre-training is beneficial, as the model trained from scratch underperforms in comparison: the 130 million parameters model trained from scratch is outperformed by a pre-trained model of less than half the size.

---

[3]GitHub repository: `https://github.com/shehzaidi/pre-training-via-denoising`.

## 5.4 PRE-TRAINING IMPROVES FORCE PREDICTION

As shown in Section 3.2.1, denoising structures corresponds to learning an approximate force field directly from equilibrium structures. We explore whether pre-training via denoising would therefore also improve models trained to predict atomic forces. We compare the performance of TorchMD-NET for force prediction on the MD17 (aspirin) dataset with and without pre-training on PCQM4Mv2. Table 3 shows that pre-training improves force prediction. We also assess the effect of pre-training on the OC20 dataset for force prediction in Appendix G.3, similarly finding an improvement due to pre-training.

Table 3: Performance of TorchMD-NET for force prediction on MD17 (aspirin).

| Method | Test MAE |
| --- | --- |
| TorchMD-NET | $0.268 \pm 0.003$ |
| + Pre-training | $\mathbf{0.222} \pm 0.003$ |

## 5.5 FREEZING PRE-TRAINED PARAMETERS

Finally, we perform an experiment to assess how useful the features learned by pre-training are if they are not fine-tuned for the downstream task but kept fixed instead. Specifically, on the HOMO target in QM9, we freeze the backbone of the model and fine-tune only the decoder (*cf.* Appendix A). To evaluate this, we compare it to using random parameters from initialization for the model's backbone, which allows us to isolate how useful the pre-trained features are. As described in Appendix A, the decoder is a simple module involving no message-passing. Figure 5 shows that only training the decoder while keeping the pre-trained parameters fixed results in test MAE of 40 meV, which is worse than fine-tuning the entire model but substantially better than the performance of >100 meV in test MAE resulting from training the decoder when the remaining parameters are randomly initialized. This suggests that the features learned by denoising are more discriminative for downstream prediction than random features. We note that training only the decoder is also substantially faster than training the entire network – one batch on a single V100 GPU takes 15ms, which is $50\times$ faster than one batch using 16 TPUs for the full network.

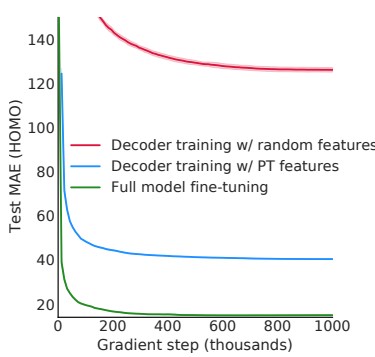

Figure 5: Training only the decoder results in significantly better performance with pre-trained instead of random features with GNS-TAT.

## 6 LIMITATIONS & FUTURE WORK

We have shown that pre-training can significantly improve performance for various tasks. One additional advantage of pre-trained models is that they can be shared in the community, allowing practitioners to fine-tune models on their datasets. However, unlike vision and NLP, molecular networks vary widely and the community has not yet settled on a "standard" architecture, making pre-trained weights less reusable. Moreover, the success of pre-training inevitably depends on the relationship between the upstream and downstream datasets. In the context of molecular property prediction, understanding what aspects of the upstream data distribution must match the downstream data distribution for transfer is an important direction for future work. More generally, pre-training models on large datasets incurs a computational cost. However, our results show that pre-training for 3D molecular prediction does not require the same scale as large NLP and vision models. We discuss considerations on the use of compute and broader impact in Appendix C.

## 7 CONCLUSION

We investigated pre-training neural networks by denoising in the space of 3D molecular structures. We showed that denoising in this context is equivalent to learning a force field, motivating its ability to learn useful representations and shedding light on successful applications of denoising in other works (Godwin et al., 2022). This technique enabled us to utilize existing large datasets of 3D structures for improving performance on various downstream molecular property prediction tasks, setting a new SOTA in some cases such as QM9. More broadly, this bridges the gap between the utility of pre-training in vision/NLP and molecular property prediction from structures. We hope that this approach will be particularly impactful for applications of deep learning to scientific problems.

## REPRODUCIBILITY STATEMENT

Our theoretical results, complete with assumptions and proofs, are included in Section 3.2.1 and Appendix B. The code to reproduce our experiments on the TorchMD-NET architecture are open source. We report hardware setup with training times in Appendix E and detailed hyperparameter settings in Appendix F. Datasets are described with their respective licenses in Appendix D.

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

# A  ARCHITECTURAL DETAILS

## A.1  STANDARD GNS

As our base model architecture we chose a Graph Net Simulator (GNS) (Sanchez-Gonzalez et al., 2020), which consists of an ENCODER which constructs a graph representation from the input $S$, a PROCESSOR of repeated message passing blocks that update the latent graph representation, and a DECODER which produces predictions. Our implementation follows Godwin et al.'s [2022] modifications to enable molecular and graph-level property predictions which has been shown to achieve strong results across different molecular prediction tasks without relying on problem-specific features.

In the ENCODER, we represent the set of atoms $S = \{(a_1, \mathbf{p}_1), (a_2, \mathbf{p}_2), \ldots, (a_{|S|}, \mathbf{p}_{|S|})\}$ as a directed graph $G = (V, E)$ where $V = \{\mathbf{v}_1, \mathbf{v}_2, \ldots, \mathbf{v}_{|S|}\}$ and $E = \{\mathbf{e}_{i,j}\}_{i,j}$ are the sets of "featurized" vertices and edges, respectively. Edges $\mathbf{e}_{i,j} \in E$ are constructed whenever the distance between the $i$-th and $j$-th atoms is less than the connectivity radius $R_{\mathrm{cut}}$ (given in Appendix F), in which case we connect $\mathbf{v}_i$ and $\mathbf{v}_j$ with a *directed* edge $\mathbf{e}_{i,j}$ from $i$ to $j$ that is a featurization of the displacement vector $\mathbf{p}_j - \mathbf{p}_i$. Meanwhile, for the $i$-th atom, $\mathbf{v}_i$ is given by a learnable vector embedding of the atomic number $a_i$.

The PROCESSOR consists of $L$ message-passing steps that produce intermediate graphs $G_1, \ldots, G_L$ (with the same connectivity structure as the initial one). Each of these steps computes the sum of a shortcut connection from the previous graph, and the application of an Interaction Network (Battaglia et al., 2016). Interaction Networks first update each edge feature by applying an "edge update function" to a combination of the existing feature and the features of the two connected vertices. They then update each vertex feature by applying a "vertex update function" to a combination of the existing feature and the (new) edge features of incoming edges. In GNS, edge update functions are 3 hidden layer fully-connected MLPs, using a "shifted softplus" ($\mathrm{ssp}(x) = \log(0.5e^x + 0.5)$) activation function, applied to the concatenation of the relevant edge and vertex features, followed by a layer normalization layer. Vertex update functions are similar, but are applied to the concatenation of the relevant vertex feature and *sum* over relevant edge features.

In our implementation of GNS we applied the same PROCESSOR in sequence three times (with shared parameters), with the output of each being decoded to produce a prediction and corresponding loss value. The loss for the whole model is then given by the average of these. (Test-time predictions are meanwhile computed using only the output of the final PROCESSOR.)

The DECODER is responsible for computing graph-level and vertex-level predictions from the output of each PROCESSOR. Vertex-level predictions, such as noise as described in Section 3.2, are decoded using an MLP applied to each vertex feature. Graph-level predictions (*e.g.* energies) are produced by applying an MLP to each vertex feature, aggregating the result over vertices (via a sum), and then applying another MLP to the result.

## A.2 GNS with Tailored Activation Transformation (GNS-TAT)

Figure 6: Diagram showing the edge update for a single step $t$ of the PROCESSOR. **Left:** Edge update for GNS. **Right:** Edge update for GNS-TAT (with modifications shown in red).

Tailored Activation Transformation (TAT) (Zhang et al., 2022) is a method for initializing and transforming neural networks to make them easier to train, and is based on a similar method called Deep Kernel Shaping (DKS) (Martens et al., 2021). TAT controls the propagation of "q values", which are initialization-time approximations to dimension-normalized squared norms of the network's layer-wise activation vectors, and "c values", which are cosine similarities between such vectors (for different inputs). In other words, q values approximate $\|z(x)\|^2/\dim(z(x))$, where $z(x)$ denotes a layer's output as a function of the network's input $x$, and c values approximate $z(x)^\top z(x')/(\|z(x)\|\|z(x')\|)$, where $x'$ is another possible network input. In standard deep networks, c values will converge to a constant value $c_\infty \in [0, 1]$, so that "geometric information" is lost, which leads to training difficulties (Martens et al., 2021). DKS/TAT prevents this convergence through a combination of careful weight initialization, and transformations to the network's activation functions and sum/average layers.

Oversmoothing (Chen et al., 2019; Cai & Wang, 2020; Rong et al., 2019; Zhou et al., 2020; Yang et al., 2020; Zhao & Akoglu, 2020; Do et al., 2021) is a phenomenon observed in GNN architectures where vertex/edge features all converge to approximately the same value with depth, and is associated with training difficulties. It is reminiscent of how, when $c_\infty = 1$, feature vectors will converge with depth to a constant input-independent vector in standard deep networks. It therefore seems plausible that applying TAT to GNNs may help with the oversmoothing problem and thus improve training performance.

Unfortunately, the GNS architecture violates two key assumptions of TAT. Firstly, the sums over edge features (performed in the vertex update functions) violate the assumption that all sum operations must be between the outputs of linear layers with *independently sampled* initial weights. Secondly, GNS networks have multiple inputs for which information needs to be independently preserved and propagated to the output, while DKS/TAT assumes a single input (or multiple inputs whose representations evolve independently in the network).

To address these issues we introduce a new initialization scheme called "Edge-Delta", which initializes to zero the weights that multiply incoming vertex features in the edge update functions (and treats these weights as absent for the purpose of computing the initial weight variance). This approach is inspired by the use of the "Delta initialization" (Balduzzi et al., 2017; Xiao et al., 2018) for convolutional networks in DKS/TAT, which initializes filter weights of the non-central locations to zero, thus allowing geometric information, in the form of c values, to propagate independently for each location in the feature map. When using the Edge-Delta initialization, edge features propagate independently of each other (and of vertex features), through what is essentially a standard deep

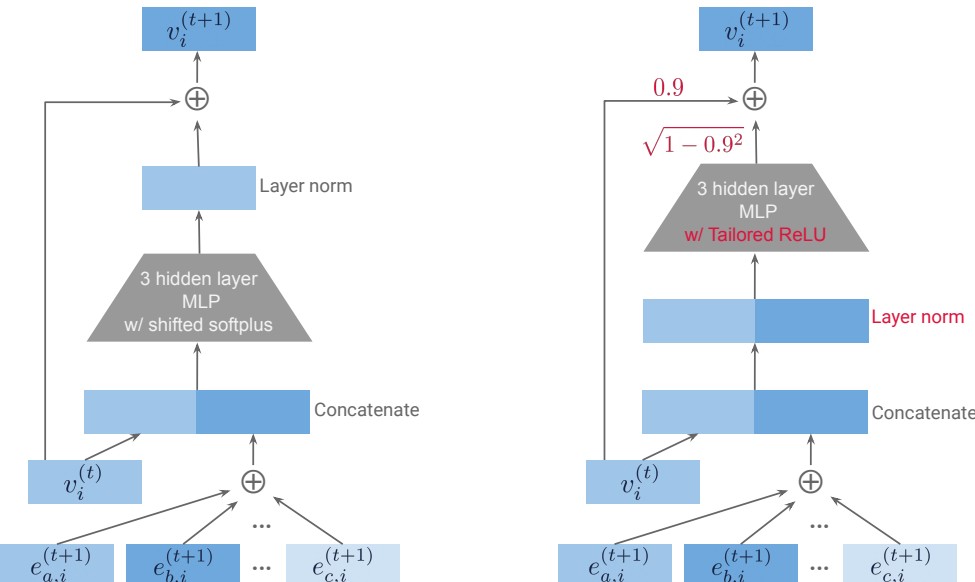

Figure 7: Diagram showing the vertex update for a single step $t$ of the PROCESSOR. **Left:** Vertex update for GNS. **Right:** Vertex update for GNS-TAT (with modifications shown in red).

residual network (with edge update functions acting as the residual branches), which we will refer to as the "edge network".

Given the use of Edge-Delta we can then apply TAT to GNS as follows[4]. First, we replace GNS's activation functions with TAT's transformed Leaky-ReLU activation functions (or "Tailored ReLUs"), which we compute with TAT's $\eta$ parameter set to $0.8$, and its "subnetwork maximizing function" defined on the edge network[5]. We also replace each sum involving shortcut connections with weighted sums, whose weights are $0.9$ and $\sqrt{1 - 0.9^2}$ for the shortcut and non-shortcut branches respectively. We retain the use of layer normalization layers in the edge/vertex update functions, but move them to before the first fully-connected layer, as this seems to give the best performance. As required by TAT, we use a standard Gaussian fan-in initialization for the weights, and a zero initialization for the biases, with Edge-Delta used only for the first linear layer of the edge update functions. Finally, we replace the sum used to aggregate vertex features in the DECODER with an average. See Figures 6 and 7 for an illustration of these changes.

We experimented with an analogous "Vertex-Delta" initialization, which initializes to zero weights in the vertex update functions that multiply summed edge features, but found that Edge-Delta gave the best results. This might be because the edge features, which encode distances between vertices (and are best preserved with the Edge-Delta approach), are generally much more informative than the vertex features in molecular property prediction tasks. We also ran informal ablation studies, and found that each of our changes to the original GNS model contributed to improved results, with the use of Edge-Delta and weighted shortcut sums being especially important.

**GNS vs. GNS-TAT on OC20.** Our preliminary experiments used to determine appropriate choices for the use of Edge-Delta/Vertex-Delta initialization, weighting for shortcut sums and TAT hyperparameters such as $\eta$ were conducted on QM9. This led to a parameterization of GNS-TAT which gave improved performance on QM9/DES15K as shown earlier in Sections 4.2 and 4.4. However, we kept

---

[4]Note that Edge-Delta initialization is compatible with TAT, since for the purposes of q/c value propagation, zero-initialized connections in the network can be treated as absent.

[5]For the purposes of computing the subnetwork maximizing function we ignore the rest of the network and just consider the edge network. While the layer normalization layer (which we move before the MLP) technically depends on the vertex features, this dependency can be ignored as long as the q values of these features is 1 (which will be true given the complete set of changes we make to the GNS architecture).

the original parameterization for GNS from Godwin et al. (2022) on OC20, as the model size was different from QM9/DES15K (see Appendix F) and re-tuning hyperparameters on OC20 would be computationally expensive. Using the same TAT settings as QM9 on OC20 results in approximately the same performance as the original GNS model as shown in Table 4.

Table 4: GNS vs. GNS-TAT energy prediction MAE on OC20. TAT hyperparameters were tuned with experiments on QM9 and then kept fixed for OC20.

| Validation Dataset | Model | | |
| --- | --- | --- | --- |
| | GNS | GNS + NN | GNS-TAT + NN |
| ID | 0.5233 | 0.4196 | 0.4199 |
| OOD Adsorbate | 0.6295 | 0.4900 | 0.5013 |
| OOD Catalyst | 0.5202 | 0.4316 | 0.4319 |
| OOD Both | 0.5617 | 0.4282 | 0.4373 |

## B  DENOISING AS LEARNING A FORCE FIELD

We specify a molecular structure as $\mathbf{x} = (\mathbf{x}^{(1)}, \ldots, \mathbf{x}^{(N)}) \in \mathbb{R}^{3N}$, where $\mathbf{x}^{(i)} \in \mathbb{R}^3$ is the coordinate of atom $i$. Let $E(\mathbf{x})$ denote the total (potential) energy of $\mathbf{x}$, such that $-\nabla_{\mathbf{x}} E(\mathbf{x})$ are the forces on the atoms. As discussed in Section 3.2.1, learning the force field, *i.e.* the mapping $\mathbf{x} \mapsto -\nabla_{\mathbf{x}} E(\mathbf{x})$, is a reasonable pre-training objective. Furthermore, learning the force field can be viewed as score-matching if we define the distribution $p_{\text{physical}}(\mathbf{x}) \propto \exp(-E(\mathbf{x}))$ and observe that the *score* of $p_{\text{physical}}$ is the force field: $\nabla_{\mathbf{x}} \log p_{\text{physical}}(\mathbf{x}) = -\nabla_{\mathbf{x}} E(\mathbf{x})$.

However, a technical caveat is that $p_{\text{physical}}$ is an *improper* probability density, because it cannot be normalized due to the translation invariance of $E$. Writing the translation of a structure as $\mathbf{x} + \mathbf{t} := (\mathbf{x}^{(1)} + \mathbf{t}, \ldots, \mathbf{x}^{(N)} + \mathbf{t})$ where $\mathbf{t} \in \mathbb{R}^3$ is a constant vector, we have $E(\mathbf{x} + \mathbf{t}) = E(\mathbf{x})$. This implies that the normalizing constant $\int_{\mathbb{R}^{3N}} p_{\text{physical}}(\mathbf{x}) \, \mathrm{d}\mathbf{x}$ diverges to infinity. To remedy this, we can restrict ourselves to the $(3N-3)$-dimensional subspace $V := \{\mathbf{x} \in \mathbb{R}^{3N} \mid \sum_i \mathbf{x}^{(i)} = 0\} \subseteq \mathbb{R}^{3N}$ consisting of the mean-centered structures, over which $p_{\text{physical}}$ can be defined as a normalizable distribution.

Proceeding similarly as Section 3.2.1, let $\mathbf{x}_1, \ldots, \mathbf{x}_n \in V$ be a set of *mean-centered* equilibrium structures. For any $\tilde{\mathbf{x}} \in V$, we now approximate

$$p_{\text{physical}}(\tilde{\mathbf{x}}) \approx q_\sigma(\tilde{\mathbf{x}}) := \frac{1}{n} \sum_{i=1}^n q_\sigma(\tilde{\mathbf{x}} \mid \mathbf{x}_i),$$

where the Gaussian distributions $q_\sigma(\tilde{\mathbf{x}} \mid \mathbf{x}_i)$ are defined on $V$ as:

$$q_\sigma(\tilde{\mathbf{x}} \mid \mathbf{x}_i) = \frac{1}{(2\pi\sigma)^{(3N-3)/2}} \exp\left(-\frac{1}{2\sigma^2} \|\tilde{\mathbf{x}} - \mathbf{x}_i\|^2\right).$$

For convenience, we have expressed structures as vectors in the ambient space $\mathbb{R}^{3N}$, however they are restricted to lie in the smaller space $V$. Note that the normalizing constant accounts for the fact that $V$ is $(3N-3)$-dimensional. As before, we define $q_0(\mathbf{x}) = \frac{1}{n} \sum_{i=1}^n \delta(\mathbf{x} = \mathbf{x}_i)$ to be the empirical distribution and $q_\sigma(\tilde{\mathbf{x}}, \mathbf{x}) = q_\sigma(\tilde{\mathbf{x}} \mid \mathbf{x}) q_0(\mathbf{x})$. The score-matching objective is given by:

$$J_1(\theta) = \mathbb{E}_{q_\sigma(\tilde{\mathbf{x}})} \left[ \|\text{GNN}_\theta(\tilde{\mathbf{x}}) - \nabla_{\tilde{\mathbf{x}}} \log q_\sigma(\tilde{\mathbf{x}})\|^2 \right], \tag{5}$$

where the expectation is now over $V$. As shown by Vincent (2011), minimizing the objective above is equivalent to the minimizing the following objective:

$$J_2(\theta) = \mathbb{E}_{q_\sigma(\tilde{\mathbf{x}}, \mathbf{x})} \left[ \|\text{GNN}_\theta(\tilde{\mathbf{x}}) - \nabla_{\tilde{\mathbf{x}}} \log q_\sigma(\tilde{\mathbf{x}} \mid \mathbf{x})\|^2 \right]. \tag{6}$$

This is recognized as a denoising objective, because $\nabla_{\tilde{\mathbf{x}}} \log q_\sigma(\tilde{\mathbf{x}} \mid \mathbf{x}) = (\mathbf{x} - \tilde{\mathbf{x}})/\sigma^2$. A practical implication of this analysis is that the noise $(\mathbf{x} - \tilde{\mathbf{x}})/\sigma^2 \in V$ should be mean-centered, which is intuitive since it is impossible to predict a translational component in the noise.

We include a proof of the equivalence between Equations (5) and (6) for completeness:

**Proposition 1** (Vincent (2011)). *The minimization objectives $J_1(\theta)$ and $J_2(\theta)$ are equivalent.*

*Proof.* We first observe:

$$J_1(\theta) = \mathbb{E}_{q_\sigma(\tilde{\mathbf{x}})}\left[\|\text{GNN}_\theta(\tilde{\mathbf{x}})\|^2\right] - 2\mathbb{E}_{q_\sigma(\tilde{\mathbf{x}})}\left[\langle\text{GNN}_\theta(\tilde{\mathbf{x}}), \nabla_{\tilde{\mathbf{x}}}\log q_\sigma(\tilde{\mathbf{x}})\rangle\right] + C_1$$

$$J_2(\theta) = \mathbb{E}_{q_\sigma(\tilde{\mathbf{x}})}\left[\|\text{GNN}_\theta(\tilde{\mathbf{x}})\|^2\right] - 2\mathbb{E}_{q_\sigma(\tilde{\mathbf{x}},\mathbf{x})}\left[\langle\text{GNN}_\theta(\tilde{\mathbf{x}}), \nabla_{\tilde{\mathbf{x}}}\log q_\sigma(\tilde{\mathbf{x}}\mid\mathbf{x})\rangle\right] + C_2,$$

where $C_1, C_2$ are constants independent of $\theta$. Therefore, it suffices to show that the middle terms on the RHS are equal. Since expectations over $q_\sigma(\tilde{\mathbf{x}})$ and $q_\sigma(\tilde{\mathbf{x}}, \mathbf{x})$ are restricted to $V \subseteq \mathbb{R}^{3N}$, we apply a change of basis to write them as integrals against the $(3N-3)$-dimensional Lebesgue measure. Pick an orthonormal basis $\{\boldsymbol{v}_1, \ldots, \boldsymbol{v}_{3N-3}\} \subseteq \mathbb{R}^{3N}$ for $V$ and let $P_V = [\boldsymbol{v}_1, \ldots, \boldsymbol{v}_{3N-3}] \in \mathbb{R}^{3N \times 3(N-1)}$ be the projection matrix, so $\mathbf{z} = P_V^\mathsf{T}\mathbf{x}$ expresses a mean-centered structure $\mathbf{x}$ in terms of the coordinates of the chosen basis for $V$. Noting that $P_V$ has orthonormal columns and that it yields a bijection between $V$ and $\mathbb{R}^{3N-3}$, we calculate:

$$\mathbb{E}_{q_\sigma(\tilde{\mathbf{x}})}\left[\langle\text{GNN}_\theta(\tilde{\mathbf{x}}), \nabla_{\tilde{\mathbf{x}}}\log q_\sigma(\tilde{\mathbf{x}})\rangle\right]$$

$$= \int_{\mathbb{R}^{3N-3}} q_\sigma(P_V\tilde{\mathbf{z}})\langle\text{GNN}_\theta(P_V\tilde{\mathbf{z}}), \nabla\log q_\sigma(P_V\tilde{\mathbf{z}})\rangle\,\mathrm{d}\tilde{\mathbf{z}}$$

$$= \int_{\mathbb{R}^{3N-3}} q_\sigma(P_V\tilde{\mathbf{z}})\left\langle\text{GNN}_\theta(P_V\tilde{\mathbf{z}}), \frac{\nabla q_\sigma(P_V\tilde{\mathbf{z}})}{q_\sigma(P_V\tilde{\mathbf{z}})}\right\rangle\,\mathrm{d}\tilde{\mathbf{z}}$$

$$= \int_{\mathbb{R}^{3N-3}} \langle\text{GNN}_\theta(P_V\tilde{\mathbf{z}}), \nabla q_\sigma(P_V\tilde{\mathbf{z}})\rangle\,\mathrm{d}\tilde{\mathbf{z}}$$

$$= \int_{\mathbb{R}^{3N-3}} \left\langle\text{GNN}_\theta(P_V\tilde{\mathbf{z}}), \frac{1}{n}\sum_{i=1}^{n}\nabla q_\sigma(P_V\tilde{\mathbf{z}}\mid\mathbf{x}_i)\right\rangle\,\mathrm{d}\tilde{\mathbf{z}}$$

$$= \int_{\mathbb{R}^{3N-3}} \left\langle\text{GNN}_\theta(P_V\tilde{\mathbf{z}}), \frac{1}{n}\sum_{i=1}^{n}q_\sigma(P_V\tilde{\mathbf{z}}\mid\mathbf{x}_i)\nabla\log q_\sigma(P_V\tilde{\mathbf{z}}\mid\mathbf{x}_i)\right\rangle\,\mathrm{d}\tilde{\mathbf{z}}$$

$$= \int_{\mathbb{R}^{3N-3}} \frac{1}{n}\sum_{i=1}^{n}q_\sigma(P_V\tilde{\mathbf{z}}\mid\mathbf{x}_i)\langle\text{GNN}_\theta(P_V\tilde{\mathbf{z}}), \nabla\log q_\sigma(P_V\tilde{\mathbf{z}}\mid\mathbf{x}_i)\rangle\,\mathrm{d}\tilde{\mathbf{z}}$$

$$= \mathbb{E}_{q_\sigma(\tilde{\mathbf{x}},\mathbf{x})}\left[\langle\text{GNN}_\theta(\tilde{\mathbf{x}}), \nabla_{\tilde{\mathbf{x}}}\log q_\sigma(\tilde{\mathbf{x}}\mid\mathbf{x})\rangle\right]$$

$\square$

## B.1 Differences Between Denoising for Generative Modelling vs. Learning Forces

Denoising has recently seen widespread use as a means for generative score-based modelling (*e.g.* Song & Ermon, 2019; 2020; Ho et al., 2020) due to the equivalence between denoising and score-matching (Vincent, 2011). In this section, we elaborate on the differences between the equivalence of denoising and score-matching for generative modelling versus the equivalence of denoising and force learning. Both of these equivalences rely on the result of Vincent (2011) although with different assumptions and different aims in practice.

As background, generative score-based modelling consists of modelling a target distribution $p(\mathbf{x})$ by learning an approximation of its score function $\nabla\log p(\mathbf{x})$ using a neural network given a training set of samples $\mathbf{x}_1, \ldots \mathbf{x}_n \sim p(\mathbf{x})$. Using the learned approximation of the score function, new samples can be generated using score-based MCMC techniques, such as Langevin MCMC. Learning the score function corresponds to the optimization objective

$$\min_\theta \mathbb{E}_{p(\mathbf{x})}\left[\|\varphi_\theta(\mathbf{x}) - \nabla\log p(\mathbf{x})\|^2\right]. \tag{7}$$

Although the expectation over $p(\mathbf{x})$ can be approximated using the samples $\mathbf{x}_i$, the true score $\nabla\log p(\mathbf{x})$ is unknown, rendering the objective function intractable as is. *Denoising score-matching* (Vincent, 2011) approaches this by replacing $p(\mathbf{x})$ with an approximation that leads to a tractable

objective. We can replace $p(\mathbf{x})$ with a *noisy approximation* $q_\sigma(\tilde{\mathbf{x}}) \coloneqq \int q_\sigma(\tilde{\mathbf{x}} \mid \mathbf{x}) p(\mathbf{x}) \, \mathrm{d}\mathbf{x}$ where $q_\sigma(\tilde{\mathbf{x}} \mid \mathbf{x}) \coloneqq \mathcal{N}(\tilde{\mathbf{x}}; \mathbf{x}, \sigma^2 I)$ is the *noising distribution*. The advantage of doing so is that score-matching with $q_\sigma(\tilde{\mathbf{x}})$ is equivalent to the following tractable denoising objective by Proposition 1:

$$\min_\theta \mathbb{E}_{q_\sigma(\tilde{\mathbf{x}}|\mathbf{x})p(\mathbf{x})} \left[ \|\varphi_\theta(\tilde{\mathbf{x}}) - \nabla \log q_\sigma(\tilde{\mathbf{x}} \mid \mathbf{x})\|^2 \right].$$

However, in order for this approximation to be accurate, it is *crucial that the noise level $\sigma$ is as small as possible*, ideally zero in which case the approximation is exact. As shown by Song & Ermon (2019), choosing a value for $\sigma$ close to zero leads to unreliable learned estimates of the score away from the regions of $p(\mathbf{x})$ with high probability density. Therefore, they proposed multi-scale denoising where the model simultaneously learns to approximate the scores of $q_{\sigma_i}$ for $i = 1, \ldots L$, where $\sigma_1 > \sigma_2 > \cdots > \sigma_L$ is a sequence of increasing noise levels such that $\sigma_L$ is close to zero and $q_{\sigma_L} \approx p$. In practice, *this then leads to multi-scale denoising*.

In the context of learning forces, we first observe that the score function of the Boltzmann distribution $p(\mathbf{x}) = p_{\text{physical}}(\mathbf{x}) \propto \exp(-E(\mathbf{x}))$ is equal to the forces $\nabla \log p_{\text{physical}}(\mathbf{x}) = -\nabla E(\mathbf{x})$. Therefore, learning the forces corresponds to the following score-matching objective:

$$\min_\theta \mathbb{E}_{p_{\text{physical}}(\mathbf{x})} \left[ \|\varphi_\theta(\mathbf{x}) - \nabla \log p_{\text{physical}}(\mathbf{x})\|^2 \right].$$

Once again, the score function $\nabla \log p_{\text{physical}}(\mathbf{x})$ is unknown and must be approximated. However, unlike previously, we now have access to a set of equilibrium, energy-minimizing structures $\mathbf{x}_1, \ldots, \mathbf{x}_n$; these examples are local maxima of the distribution $p_{\text{physical}}(\mathbf{x})$ rather than samples from it as in the case before. Let $q_\sigma(\tilde{\mathbf{x}})$ denote a mixture of Gaussians with noise scale $\sigma$ centered at $\mathbf{x}_1, \ldots, \mathbf{x}_n$. In order to capture the fact that $p_{\text{physical}}(\mathbf{x})$ has local maxima at $\mathbf{x}_1, \ldots, \mathbf{x}_n$, we can approximate $p_{\text{physical}}(\mathbf{x})$ with $q_\sigma(\tilde{\mathbf{x}})$. This distribution $q_\sigma(\tilde{\mathbf{x}})$ can also be written as $q_\sigma(\tilde{\mathbf{x}}) = \int q_\sigma(\tilde{\mathbf{x}} \mid \mathbf{x}) q_0(\mathbf{x}) \, \mathrm{d}\mathbf{x}$, where $q_0(\mathbf{x}) = \frac{1}{n} \sum_{i=1}^n \delta(\mathbf{x} = \mathbf{x}_i)$, hence we can apply Proposition 1 again and reduce the optimization objective to denoising as explained in Section 3.2.1.

*In contrast with before, in this case, a desirable noise scale $\sigma$ is not zero*, since $\mathbf{x}_1, \ldots, \mathbf{x}_n$ are local maxima of $p_{\text{physical}}$ and, intuitively, the approximation should contain some non-zero mass around each local maxima. As a result, we do not use multi-scale denoising in practice. It is also worth noting that in contrast with generative modelling, our aim is to learn the score function itself (*i.e.* forces) rather than producing new samples by using the learned score function with MCMC approaches.

## C  BROADER IMPACT

**Who may benefit from this work?**  Molecular property prediction works towards a range of applications in materials design, chemistry, and drug discovery. Wider use of pre-trained models may accelerate progress in a similar manner to how pre-trained language or image models have enabled practitioners to avoid training on large datasets from scratch. Pre-training via denoising is simple to implement and can be immediately adopted to improve performance on a wide range of molecular property prediction tasks. As research converges on more standardized architectures, we expect shared pre-trained weights will become more common across the community.

**Potential negative impact and ethical considerations.**  Pre-training models on large structure datasets incurs additional computational cost when compared to training a potentially smaller model with less capacity from scratch. Environmental mitigation should be taken into account when pre-training large models (Patterson et al., 2021). However, the computational cost of pre-training can and should be offset by sharing pre-trained embeddings when possible. Moreover, in our ablations of upstream dataset sizes for GNS-TAT, we observed that training on a subset of PCQM4Mv2 was sufficient for strong downstream performance. In future work, we plan to investigate how smaller subsets with sufficient diversity can be used to minimize computational requirements, *e.g.* by requiring fewer gradient steps.

## D  DATASETS

**PCQM4Mv2.** The main dataset we use for pre-training is PCQM4Mv2 (Nakata & Shimazaki, 2017) (license: CC BY 4.0), which contains 3,378,606 organic molecules, specified by their 3D structures

at equilibrium (atom types and coordinates) calculated using DFT. Molecules in PCQM4Mv2 have around 30 atoms on average and vary in terms of their composition, with the dataset containing 22 unique elements in total. The molecules in PCQM4Mv2 only contain one label, unlike *e.g.* QM9, which contains 12 labels per molecule, however we do not use these labels as denoising only requires the structures.

**QM9.** QM9 is a dataset (Ramakrishnan et al., 2014) (license: CCBY 4.0) with approximately 130,000 small organic molecules containing up to nine heavy C, N, O, F atoms, specified by their structures. Each molecule has 12 different labels corresponding to different molecular properties, such as highest occupied molecular orbital (HOMO) energy and internal energy, which we use for fine-tuning. Following prior work, we randomly split the dataset into 114k examples for training, 10k examples for validation and 10k examples for testing.

**OC20.** Open Catalyst 2020 (Chanussot* et al., 2021) (OC20, license: CC Attribution 4.0) is a recent large benchmark containing trajectories of interacting surfaces and adsorbates that are relevant to catalyst discovery and optimization. This dataset contains three tasks: predicting the relaxed state energy from the initial structure (IS2RE), predicting the relaxed structure from the initial structure (IS2RS) and predicting the energy and forces given the structure at any point in the trajectory (S2EF). For IS2RE and IS2RS, there are 460,000 training examples, where each data point is a trajectory of a surface-adsorbate molecule pair starting with a high-energy initial structure that is relaxed towards a low-energy, equilibrium structure. For S2EF, there are 113 million examples of (non-equilibrium) structures with their associated energies and per-atom forces. We evaluate the models on the provided validation sets, some of which include out-of-distribution data relative to the training data, as detailed in Chanussot* et al. (2021).

**DES15K.** DES15K (Donchev et al., 2021) (license: CC0 1.0) is a small dataset containing around 15,000 interacting molecule pairs, specifically dimer geometries with non-covalent molecular interactions. Each pair is labelled with the associated interaction energy computed using the coupled-cluster method with single, double, and perturbative triple excitations (CCSD(T)) (Bartlett & Musiał, 2007), which is widely regarded as the gold-standard method in electronic structure theory. We randomly split the dataset into 13k training examples, 1k validation examples and 1k testing examples.

**Usage of DFT-generated structures for pre-training.** The structures in PCQM4Mv2 are obtained using DFT calculations, which *in principle* could have also been used to generate labels for molecular properties in DFT-generated downstream datasets, such as QM9. However, there are multiple reasons why denoising remains a desirable pre-training objective in such settings. First, although there is a computational cost for generating datasets such as PCQM4Mv2, it is now openly available and part of our aim is to understand how to leverage such datasets for tasks on other datasets (analogous to how ImageNet is expensive to build, but once it is available, it is important to understand how it can improve downstream performance on other datasets). Second, even if PCQM4Mv2 contained all the labels in QM9, pre-training via denoising structures allows one to pre-train a single, *label-agnostic* model which can be individually fine-tuned on any of the targets in QM9. This is substantially cheaper than per-target pre-training, and the resulting pre-trained model is also re-useable for differing needs and downstream tasks.

In other settings where the downstream dataset is generated using more expensive methods than DFT, pre-training via denoising DFT-relaxed structures can also be helpful and has a clear benefit, as shown by our experiment on the CCSD(T)-generated dataset DES15K where denoising on PCQM4Mv2 improves performance for a downstream task involving a "higher" level of theory such as CCSD(T). Generally, we emphasize that the methodology of denoising structures can be applied to any dataset of structures (regardless of whether they are computed using DFT or not). We hope that denoising will be useful in the future for learning representations by pre-training on structures obtained through other methods such as experimental data (in which case labeling may be expensive) and databases generated by other models such as AlphaFold (where only structures are available) (Jumper et al., 2021).

# E  EXPERIMENT SETUP AND COMPUTE RESOURCES

Below, we list details on our experiment setup and hardware resources used.

**GNS & GNS-TAT.** GNS-TAT training for QM9, PCQM4Mv2 and DES15K was done on a cluster of 16 TPU v3 devices and evaluation on a single V100 device. GNS training for OC20 was done on 8 TPU v4 devices, with the exception of the 1.2 billion parameters variant of the model, which was trained on 64 TPU v4 devices. Pre-training on PCQM4Mv2 was executed for $3 \cdot 10^5$ gradient updates (approximately 1.5 days of training). Fine-tuning experiments were run until convergence for QM9 ($10^6$ gradient updates taking approximately 2 days) and DES15K ($10^5$ gradient updates taking approximately 4 hours) and stopped after $5 \cdot 10^5$ gradient updates on OC20 (2.5 days) to minimize hardware use (the larger models keep benefiting from additional gradient updates).

**TorchMD-NET.** We implemented denoising for TorchMD-NET on top of Thölke & De Fabritiis's [2022] open-source code.[6] Models were trained on QM9 using data parallelism over two NVIDIA RTX 2080Ti GPUs. Pre-training on PCQM4Mv2 was done using three GPUs to accommodate the larger molecules while keeping the batch size approximately the same as QM9. All hyperparameters except the learning rate schedule were kept fixed at the defaults. Pre-training took roughly 24 hours, whereas fine-tuning took around 16 hours.

**Hyperparameter optimization.** We note that effective pre-training via denoising requires sweeping noise values, as well as loss co-efficients for denoising and atom type recovery. For GNS/GNS-TAT, we relied on the hyperparameters published by Godwin et al. (2022) but determined new noise values for pre-training and fine-tuning by tuning over the set of values $\{0.005, 0.01, 0.02, 0.05, 0.1\}$ for each of PCQM4Mv2 and QM9 (on the HOMO energy target). We used the same values for DES15K without modification. We also ran a similar number of experiments to determine cosine cycle parameters for learning rates.

---

[6]Available on GitHub at: `https://github.com/torchmd/torchmd-net`.

## F HYPERPARAMETERS

We report the main hyperparameters used for GNS and GNS-TAT below.

Table 5: GNS-TAT hyperparameters for pre-training on PCQM4Mv2.

| Parameter | Value or description |
|---|---|
| Gradient steps | $3 \cdot 10^5$ |
| Optimizer | Adam with warm up and 1-cycle cosine decay schedule |
| $\beta_1$ | 0.9 |
| $\beta_2$ | 0.95 |
| Warm up steps | $10^4$ |
| Warm up start learning rate | $10^{-5}$ |
| Warm up max learning rate | $10^{-4}$ |
| Cosine min learning rate | $10^{-7}$ |
| Cosine cycle length | $5 \cdot 10^5$ |
| Loss type | Mean squared error |
| Batch size | Dynamic to max edge/vertex/graph count |
| Max vertices in batch | 256 |
| Max edges in batch | 9216 |
| Max graphs in batch | 8 |
| Distance featurization | Bessel first kind |
| Max edges per vertex | 20 |
| $R_{\text{cut}}$ | 10 |
| MLP number of layers | 3 |
| MLP hidden sizes | 1024 |
| Activation | Tailored ReLU (with negative slope chosen using TAT) |
| message passing layers | 10 |
| Block iterations | 3 |
| Vertex/edge latent vector sizes | 512 |
| Decoder aggregation | Mean |
| Position noise | Gaussian ($\mu = 0, \sigma = 0.02$) |
| Parameter update | Exponentially moving average (EMA) smoothing |
| EMA decay | 0.9999 |
| Position loss coefficient | 1.0 |
| Atom type mask probability | 0.75 |
| Atom type loss coefficient | 4.0 |

Table 6: GNS-TAT hyperparameters for fine-tuning on QM9 and DES15K.

| Parameter | Value or description |
|---|---|
| Gradient steps | $10^6$ QM9 / $10^5$ DES15K |
| Optimizer | Adam with warm up and 1-cycle cosine decay schedule |
| $\beta_1$ | 0.9 |
| $\beta_2$ | 0.95 |
| Warm up steps | $10^4$ |
| Warm up start learning rate | $10^{-5}$ |
| Warm up max learning rate | $10^{-4}$ |
| Cosine min learning rate | $3 \cdot 10^{-7}$ |
| Cosine cycle length | $10^6$ QM9 / $10^5$ DES15K |
| Loss type | Mean squared error |
| Batch size | Dynamic to max edge/vertex/graph count |
| Max vertices in batch | 256 |
| Max edges in batch | 3072 |
| Max graphs in batch | 8 |
| Distance featurization | Bessel first kind |
| Max edges per vertex | 20 |
| $R_{\mathrm{cut}}$ | 10 |
| MLP number of layers | 3 |
| MLP hidden sizes | 1024 |
| Activation | Tailored ReLU (with negative slope chosen using TAT) |
| message passing layers | 10 |
| Block iterations | 3 |
| Vertex/edge latent vector sizes | 512 |
| Decoder aggregation | Mean |
| Position noise | Gaussian ($\mu = 0, \sigma = 0.05$) |
| Parameter update | Exponentially moving average (EMA) smoothing |
| EMA decay | 0.9999 |
| Position loss coefficient | 0.01 |
| Atom type mask probability | 0.0 |
| Atom type loss coefficient | 0.0 |

Table 7: GNS hyperparameters for OC20.

| Parameter | Value or description |
|---|---|
| Gradient steps | $5 \cdot 10^5$ |
| Optimizer | Adam with warm up and 1-cycle cosine decay schedule |
| $\beta_1$ | 0.9 |
| $\beta_2$ | 0.95 |
| Warm up steps | $5 \cdot 10^5$ |
| Warm up start learning rate | $10^{-5}$ |
| Warm up max learning rate | $10^{-4}$ |
| Cosine min learning rate | $5 \cdot 10^{-6}$ |
| Cosine cycle length | $5 \cdot 10^6$ |
| Loss type | Mean squared error |
| Batch size | Dynamic to max edge/vertex/graph count |
| Max vertices in batch | 1024 |
| Max edges in batch | 12800 |
| Max graphs in batch | 10 |
| Distance featurization | Gaussian ($\mu = 0, \sigma = 0.5$) |
| Max edges per vertex | 20 |
| $R_{\text{cut}}$ | 6 |
| MLP number of layers | 3 |
| MLP hidden sizes | 1024 |
| Activation | shifted softplus |
| message passing layers | 5 |
| Block iterations | 5 |
| Vertex/edge latent vector sizes | 512 |
| Decoder aggregation | Sum |
| Position noise | Gaussian ($\mu = 0, \sigma = 0.2$) |
| Parameter update | Exponentially moving average (EMA) smoothing |
| EMA decay | 0.9999 |
| Position loss coefficient | 1.0 |
| Atom type mask probability | 0.0 |
| Atom type loss coefficient | 0.0 |

# G  ADDITIONAL EXPERIMENTAL RESULTS

## G.1  PERFORMANCE OF TORCHMD-NET ON DES15K

In addition to the experiments involving GNS-TAT on DES15K in Section 4.4, we also consider the performance of TorchMD-NET on DES15K with and without pre-training on PCQM4Mv2. As shown in Table 8, TorchMD-NET outperforms GNS-TAT when trained from scratch, and pre-training then yields a further boost in performance as with GNS-TAT.

Table 8: Performance of TorchMD-NET with and without pre-training for interaction energy prediction on DES15K.

| Model | Test MAE (kcal/mol) |
|---|---|
| TorchMD-NET | 0.721 |
| + Pre-training on PCQM4Mv2 | **0.406** |

## G.2    COMPARISON OF OC20 IS2RE PRE-TRAINING PERFORMANCE WITH OTHER ARCHITECTURES

Table 9 compares the performance of our models with various other architectures proposed in prior work. GNS yields SOTA performance on each of the four validation sets for the direct IS2RE task. As discussed in Section 4.3 and shown in Figure 3 (left), the model pre-trained on OC20 itself achieves SOTA performance with faster convergence than all other GNS variants. Note that since we pre-train on OC20 itself, the pre-trained model performs equally well at convergence as the model trained from scratch with noisy nodes (*cf.* Section 4.3).

Table 9: Comparison of different variants of GNS with other baseline architectures on IS2RE prediction for OC20.

| Model | Validation MAE for IS2RE | | | | |
| | ID | OOD Adsorbate | OOD Catalyst | OOD Both | Average |
|---|---|---|---|---|---|
| DimeNet ++ | 0.5636 | 0.7127 | 0.5612 | 0.6492 | 0.6217 |
| GemNet | 0.5561 | 0.7342 | 0.5659 | 0.6964 | 0.6382 |
| SphereNet | 0.5632 | 0.6682 | 0.5590 | 0.6190 | 0.6024 |
| SEGNN | 0.5310 | 0.6432 | 0.5341 | 0.5777 | 0.5715 |
| GNS | 0.5233 | 0.6295 | 0.5202 | 0.5617 | 0.5587 |
| GNS + NN | 0.4196 | 0.4900 | 0.4316 | 0.4282 | 0.4424 |
| GNS + NN (PT on PCQM4Mv2) | **0.4135** | 0.4856 | **0.4245** | 0.4245 | **0.4370** |
| GNS + NN (PT on OC20) | 0.4164 | **0.4836** | 0.4267 | **0.4237** | 0.4376 |

## G.3    PRE-TRAINING VIA DENOISING FOR FORCE PREDICTION ON OC20

Recall that in Section 5.4 we explored whether pre-training also improves force prediction models, given the link between denoising and learning forces as described in Section 3.2.1. In this section, we consider a second experiment for force models. The OC20 dataset contains a force prediction task (S2EF), where a model is trained to predict point-wise energy and forces (each point being a single DFT evaluation during a relaxation trajectory) from a given 3D structure. Tables 10 and 11 show the performance of GNS when trained from scratch vs. pre-trained via denoising on equilibrium structures in OC20, as described in Section 4.3. We show two metrics for measuring force prediction performance on each of the four validation datasets: mean absolute error (lower is better) and cosine similarity (higher is better). We observe that the pre-trained model improves upon the model trained from scratch for both metrics and all four validation datasets, with improvements up to 15%.

Table 10: Force prediction on OC20 by MAE (lower is better).

| Validation Dataset | Model | |
| | GNS | Pre-trained GNS |
|---|---|---|
| ID | 0.0332 | **0.0282** |
| OOD Adsorbate | 0.0366 | **0.0314** |
| OOD Catalyst | 0.0360 | **0.0335** |
| OOD Both | 0.0406 | **0.0382** |

Table 11: Force prediction on OC20 by cosine similarity (higher is better).

| Validation Dataset | Model | |
| | GNS | Pre-trained GNS |
|---|---|---|
| ID | 0.4845 | **0.5517** |
| OOD Adsorbate | 0.4730 | **0.5414** |
| OOD Catalyst | 0.4553 | **0.4983** |
| OOD Both | 0.4417 | **0.4849** |

