# OpenReview forum: "Pre-training via Denoising for Molecular Property Prediction"
_ICLR.cc/2023/Conference — ICLR 2023 notable top 25%_

### Official Review · Reviewer_tMrH · 2022-10-21

**Confidence:** 5
**Correctness:** 3
**Technical Novelty And Significance:** 3
**Empirical Novelty And Significance:** 3
**Recommendation:** 6

**Clarity, Quality, Novelty And Reproducibility:**

The paper is very clear to read. The problem being solved, i.e., 3D molecular representation pretraining, is novel and significant. The proposed method is not very novel, but this is okay given importance of the problem. I also think the proposed method being simple is very useful for being applied to practice.

**Strength And Weaknesses:**

Strength 1: The problem of pretraining 3D molecular representation using a small number of labels seems very significant.

Strength 2: The proposed denoising objective is quite simple and works well across varying datasets.

Strength 3: Tailored activation transformation is also useful trick for improving 3D GNN performance in general.

Weakness 1: The experiment does not consider applying the denoising objective for the recently proposed 3D graph neural network architectures like GemNet, SphereNet, or SEGNN. It is unclear whether if the proposed approach is useful for the recently proposed architectures too.

Weakness 2: The interpretation of denoising objective as a score matching objective is not very novel.

Weakness 3: The experiments do not compare with the recently proposed non-3D molecular representation learning frameworks. While attribute masking is considered as a baseline, there exist more recently proposed works, like GraphMVP or 3D infomax, which considered 3D information during training.

Weakness 4: While I appreciate how the authors provide a detailed description of their hyperparameter in Appendix E and F, the description could be made more clear for the future works to build on. For example, in Appendix E, it is unclear what the authors mean by tuning on “approximately” 5 values (hyperparameters) on the QM9 dataset. What do the authors mean by approximate? Also, is the validation set used for tuning the hyperparameters? I hope the authors clearly describe this since future works will likely follow similar protocols.


**Summary Of The Paper:**

This paper investigates (1) the denoising objective for pretraining and (2) tailored activation transformation (TAT) for 3D graph neural networks. The authors show how their denoising objective is equivalent to learning molecular force field based on interpretation of denoising objective as a score function. The proposed approach demonstrates a notable improvement over several tasks.

**Summary Of The Review:**

I recommend acceptance for this work given the following strengths:
- significance of the problem being solved
- the proposed methodology being simple and easy to implement
- well-executed experiments demonstrating benefit of the proposed approach over various settings

---

> ### Author Response · Authors · 2022-11-17
> **Response to Reviewer tMrH**
>
> Thank you very much for your review and constructive feedback! We respond to each of your points below:
> 1. **Applying the proposed approach to recently proposed architectures.** Our experiments on TorchMD-NET were aimed at demonstrating that pre-training via denoising is useful for recently proposed architectures other than GNS (which itself was originally proposed by Sanchez-Gonzalez et al. (2020), but adapted for molecular tasks recently by Godwin et al. (2022)). Note that TorchMD-NET is very recent work by Thölke & De Fabritiis (2022) [1], having been proposed earlier this year at ICLR 2022. We chose TorchMD-NET for our experiments in Section 5.1, because it is recent and has been shown to perform well compared to other baselines, yielding SOTA results. It outperforms both SEGNN and SphereNet on QM9. Moreover, it is a transformer-based architecture, as opposed to a GNN-based one like GNS (so our experiments cover two broad families of models), and it is built specifically for 3D molecules with the appropriate equivariance properties. As discussed in our related work section, pre-training via denoising is architecture-agnostic, and we conjecture that it will improve performance on architectures other than TorchMD-NET and GNS as well. A recent follow-up work [2] has already shown its effectiveness for another architecture inspired by Graphormer.
> 2. **Novelty of the link between denoising/score-matching and learning force fields.** The link between denoising and score-matching is well-known – as cited in our text, it was first shown by Vincent (2011) and then used for generative modeling by Song & Ermon (2019) and others. However, a core part of our contribution is making the connection between learning a force field (an important concept in molecular dynamics and computational chemistry) and denoising approaches in machine learning (e.g. score-based modelling). This link explains why the denoising mechanism is natural for representation learning in the context of molecules.
> There are also some mathematical differences between the denoising 🡘 score-matching equivalence for generative modelling and the denoising 🡘 learning forces equivalence. We have added a detailed discussion of these differences, along with the necessary background, in Appendix B.1 of the revised manuscript, which is summarized as follows. In denoising score-matching for generative modeling, the goal is to use samples from the target distribution to learn its score function with the end goal of generating new samples. However, in force learning, we assume access to only local maxima of the target distribution (i.e. equilibrium, energy-minimizing conformations that are local maxima of the Boltzmann distribution). As detailed in Appendix B.1, this implies that the desired noise scale in the resulting noisy approximation should be greater than zero for force learning but ideally close to zero for denoising score-matching in generative modelling. Practically, score-based generative modelling therefore requires multi-scale denoising (Song & Ermon, 2019), whereas we rely on a single noise scale. We will expand Section 3.2.1 to incorporate this discussion (currently in Appendix B.1) into the main text if accepted.
> 3. **Comparison to non-3D baselines.** The focus of our work was on 3D molecular representation learning, in part due to a lack of such methods in current literature, hence our experiments were aiming at demonstrating the method’s effectiveness for 3D tasks such as QM9. Generally, non-3D methods are not designed to handle 3D-based downstream tasks such as those that we benchmarked on. However, we anticipate that denoising can be adapted to hybrid settings containing both “2D” and 3D information. The aforementioned follow-up work [2] has shown that pre-training via denoising can be combined with other non-3D pre-training methods to learn a joint model that outperforms many previous models, which is an exciting avenue for future work.
> 4. **Tuning hyperparameters on QM9.** Thank you for pointing this out! We tuned the noise scale ($\sigma$ in Section 3.2.1) for denoising over the following grid: {0.005, 0.01, 0.02, 0.05, 0.1}. The validation set was used for hyperparameter tuning as well as early stopping. We have clarified this in Appendix E now – thank you!
>
> Thank you again for your review and suggestions to improve the paper! We hope that we have addressed your concerns, and please let us know if you have any further questions.
>
> [1] *TorchMD-NET: Equivariant Transformers for Neural Network based Molecular Potentials.* Thölke & De Fabritiis, ICLR 2022.
> [2] *One Transformer Can Understand Both 2D & 3D Molecular Data.* Luo et al., arXiv 2022. (https://arxiv.org/abs/2210.01765)

---

> > ### Comment · Reviewer_tMrH · 2022-11-17
> > **Thank you for the detailed response!**
> >
> > The authors have resolved my concerns and I remain positive about this paper.

---

### Official Review · Reviewer_YJNn · 2022-10-21

**Confidence:** 4
**Correctness:** 4
**Technical Novelty And Significance:** 4
**Empirical Novelty And Significance:** 2
**Recommendation:** 8

**Clarity, Quality, Novelty And Reproducibility:**

Although the paper is well-written, there are a few areas which could be improved in terms of clarity:
- Since many of the analyses focus on different models (e.g. GNS, GNS-TAT, GNS-TAT + PT, etc.), it would be helpful to clearly list the model being used in each table, figure, and caption
- In Fig. 3 (Left), the shading around the black line is green, which is confusing and I suspect should have been gray
- $R_{cut}$ is not defined anywhere in the paper
- Bolding the best numbers would be helpful in Table 8

**Strength And Weaknesses:**

Overall, the paper demonstrates a solid contribution. It provides several compelling analyses which show how the proposed method can generate embeddings which assist in several downstream tasks. The authors compare their method to many existing architectures, and even show that pretraining can assist another architecture (which is compatible with their proposed method). The paper also analyzes a few design choices (e.g. model size and dataset size) to show the effect of these choices on the model’s performance.

There are a few areas which I feel are weaker:
### Limited comparisons with other models from prior work
The comparisons with prior work that are shown in the paper are good to see (i.e. Table 1 and Table 8), as they directly compare GNS and/or GNS-TAT with the performance of other models. However, Table 1 only shows the comparison with other models on a single benchmark, QM9. The improvements brought on by GNS-TAT + PT + NN are certainly there, but not ubiquitous and not always huge. The comparisons in Table 8 notably do not include models such as TorchMD-NET. All other analyses in the paper are effectively only comparing GNS/GNS-TAT with itself (when PT/NN are added), or when other models like TorchMD-NET are pretrained using this method. This suggests that although the improvements in performance offered by this paper are there, they may be limited to certain datasets/benchmarks, and are not necessarily as strong. Since one of the main claims made by the paper is that PT/NN/TAT help performance, it is important to directly compare the performance of GNS/GNS-TAT (with PT/NN) with all other models (especially the strongest ones like TorchMD-NET) on more than just one benchmark. Of course, showing that TorchMD-NET performs better with pretraining (which is already done in a limited capacity) is also helpful to expand upon.
### There are several models being analyzed
Most analyses seem to use either GNS or GNS-TAT, and many of the analyses add either PT or NN (or both). Unfortunately, some analyses use GNS-TAT and others use GNS (with different flavors of adding PT or NN). This can be a little confusing for the reader, and also casts a bit of doubt on the robustness of the method. It is very possible that some benchmarks/tasks do better with TAT and others do not (or with or without NN, etc.), and if so, this should be stated explicitly as a potential caveat (i.e. TAT or NN may be better suited for some tasks and may be worse for others).
### Force-field prediction is expected to be significantly better
The connection between the denoising objective and Boltzmann-distribution force fields is a good insight offered, and it certainly suggests that the model should yield large improvements in force-field prediction. However, the main text only offers one analysis on this, which is that pretraining on TorchMD-NET seems to improve the MAE of the predicted force field for a single molecule, aspirin. Appendix G offers some more results in improving force-field predictions comparing GNS with and without pretraining. The paper would be stronger if these results were included in the main text, and it would be even better if the authors could show significant improvements in force-field prediction with GNS (or GNS-TAT) with pretraining, similar to Table 1.

**Summary Of The Paper:**

The paper proposes a pretraining method to encode 3D small molecules into atom-level encodings which can be useful for downstream prediction tasks. The method is essentially to start with the 3D coordinates of a molecule, add random Gaussian noise to the atom coordinates, and train a 3D GNN to predict the added noise. The authors connect this method to denoising score matching, which has been proposed previously in the context of diffusion models. The authors also offer an interpretation of the method as a way to approximate the local force field of the atoms, which the GNN is attempting to predict. The paper demonstrates that pretraining using this method (along with a few other tricks like Noisy Nodes (NN) and Tailored Activation Transformation (TAT)) can be used in conjunction with pretraining to achieve better predictive performance on a variety of downstream tasks.

**Summary Of The Review:**

This paper presents a novel method of pretraining for small molecules, which has some nice theoretical connections to denoising score matching, and has a direct connection with statistical mechanics. The authors show several compelling analyses which demonstrate the effectiveness of pretraining in a variety of situations. The improvements over current SOTA are perhaps modest, but present. The authors also explore the effects of pretraining combined with other techniques such as Noisy Nodes, and an application of Tailored Activation Transformation to GNNs. The analyses are limited in that there are very few direct comparisons of their GNNs with pretraining to current SOTA models, certain analyses focus on different GNNs for unknown/unjustified reasons, and the task which pretraining should theoretically benefit the most does not seem to enjoy these benefits as significantly as one would hope. Regardless, I believe the authors have shown that their method of pretraining is functional and works well, and their numerous analyses on their method help give the reader intuition on how and why pretraining, Noisy Nodes, and Tailored Activation Transformation can be helpful for molecular prediction tasks.

---

> ### Author Response · Authors · 2022-11-17
> **Response to Reviewer YJNn**
>
> Thank you very much for your thorough feedback and positive comments! We respond to each of your points below:
> 1. **“Limited comparisons with other models from prior work.”** One of the main aims of our experiments was to show that pre-training via denoising is effective, i.e. performance with pre-training is better than training from scratch. We also aimed to show that pre-training can lead to models that outperform existing models in the literature in terms of absolute performance. As you mentioned, Tables 1 and 8 (which is now 9) therefore compare the performance of our models both with/without pre-training and versus other existing models in the literature on the QM9 and OC20 benchmarks. Generally, the results show improvements over the previous best models across multiple tasks/datasets, with the amount of improvement varying but in multiple cases, it is as large as a 20-25% reduction in test loss. As you suggested, we will move Table 9 into the main paper if accepted. (Currently, it’s in the appendix due to space constraints, but we agree that it should be in the main text.) TorchMD-NET is not in Table 9 because it has not been benchmarked on OC20 to our knowledge (their open source code also does not support OC20), and training it on OC20 with sufficient hyperparameter tuning (the architecture in particular) is computationally expensive due to the size of OC20. Nonetheless, since the pre-training method is architecture-agnostic, we would expect that a given model can likely be further improved if pre-trained (as shown for TorchMD-NET in Section 5.1 for example). We also note that recent follow-up work [1] has already shown the effectiveness of pre-training via denoising – for some tasks, surpassing the performance of all previous models – for another architecture, inspired by Graphormer.
> 2. **“There are several models being analyzed.”** Thank you for this point! As you correctly note, our experiments on QM9 and DES15K used GNS-TAT, whereas on OC20 we used GNS. Our research on TAT was conducted on QM9 (as it is computationally cheaper than OC20). Our preliminary experiments were used to determine appropriate choices for the use of Edge-Delta/Vertex-Delta initialization, weighting for skip connections, and DKS/TAT hyperparameters such as $\eta$. This led to hyperparameters for GNS-TAT which gave improved performance on QM9/DES15K. However, we kept the original parameterization of GNS from Godwin et al. (2022) on OC20, as the model size was different from QM9/DES15K (see Tables 6 and 7) and re-tuning the hyperparameters on OC20 is expensive. We have added new experimental results (Table 4 in Appendix A.2) showing that using the same TAT settings as QM9 on OC20 results in almost the same performance as GNS on OC20. As suggested, we have added this discussion to the paper in Appendix A.2 to improve clarity for future readers and also improved the labels in the figures to ensure the model is clear – thank you again!
> 3. **“Force-field prediction is expected to be significantly better.”** As suggested, we will expand Section 5.4 and move the force experiments in Tables 10 and 11 from the appendix into the main text if accepted. In these experiments, we observed improved force prediction performance on MD17 and OC20 when pre-training. As discussed in Section 6, we certainly believe that identifying the settings which maximize transfer from pre-training to downstream tasks such as force prediction is an important direction for future work. This includes both picking the “right” pre-training dataset and choosing a noise scale that ensures the local approximation of the force field around equilibrium structures learned through denoising is useful for the downstream task.
> 4. **General improvements for clarity.** Thank you for your suggestions relating to clarity – all have been incorporated into the manuscript. Note that Fig. 3 (left) has a dark green line, hence the shading around it is also dark green. We changed the colors slightly to make it more prominent. We have also described $R_\text{cut}$ in Appendix A.1 and added its values in Tables 5-7.
>
> Thank you again for your feedback and suggestions – we hope that we have addressed your questions. Please let us know if you have any further questions.
>
> [1] *One Transformer Can Understand Both 2D & 3D Molecular Data.* Luo et al., arXiv 2022 (https://arxiv.org/abs/2210.01765)

---

### Official Review · Reviewer_8vAk · 2022-10-22

**Confidence:** 4
**Correctness:** 3
**Technical Novelty And Significance:** 3
**Empirical Novelty And Significance:** 3
**Recommendation:** 8

**Clarity, Quality, Novelty And Reproducibility:**

Clarity: The paper is well-written and easy to follow.

Quality: The proposed method, the experiment design, and the result analysis are all look good.

Novelty: Although the proposed method is simple, it is effective.

Reproducibility:  The paper provides details of the experiments, some are unclear (refer to "Weaknesses").

**Strength And Weaknesses:**

Strengths:
- The proposed method is simple and effective
- The analysis of the connection between denoising and force field is insightful
- The experiment is comprehensive, and the results are impressive.

Weaknesses & Questions:
- Downstream applications. There are many molecules that do not have DFT 3D conformations, and thus cannot use the proposed 3D denoising task. Therefore, the proposed method seems cannot be used in many traditional molecular property prediction tasks, like MoleculeNet.
- Some experiment settings are unclear, for example, the data split setting for QM9.
- In Sec.5.1, Only TorchMD-Net model is tested. Did you try some transformer-based architectures, like Graphormer?
- will you open-source the model/code?


**Summary Of The Paper:**

The paper proposes a simple and efficient pretraining strategy, which predicts the added noise into 3D positions. The paper further shows the connection between denoising and molecular force field learning. Experiments demonstrate the effectiveness of the proposed method.

**Summary Of The Review:**

Overall, I think this is a good paper, and recommend accepting it.

---

> ### Author Response · Authors · 2022-11-17
> **Response to Reviewer 8vAk**
>
> Thank you very much for your thoughtful feedback and appreciation! We respond to each of your points below:
> 1. **Downstream applications without 3D conformations.**
> Indeed, our pre-training methodology is specifically aimed at models taking 3D structures as input rather than “2D graphs”. Although methods for pre-training models based on 2D graphs exist in the literature, no methods for pre-training 3D structure-based models exist to the best of our knowledge, which was part of the motivation for our work. As you mentioned, 3D conformations are not always available, however structural information in datasets is becoming increasingly prevalent in the community. Examples include the recent AlphaFold databases, the new PCQM4Mv2 dataset and the recent DES datasets (DES15K and others), all of which provide 3D conformations. We therefore anticipate an increasing need for effective pre-training methods for 3D structure-based models.
> 2. **Data split setting for QM9.** Thank you for pointing this out! In line with prior work, we split the QM9 dataset into training, validation and testing sets uniformly at random with approximately 114k examples for training, 10k examples for validation and 10k examples for testing. We have clarified this in Appendix D now – thank you!
> 3. **Using a transformer-based architecture.** TorchMD-NET is a recent transformer-based architecture (introduced in “*TorchMD-NET: Equivariant Transformers for Neural Network based Molecular Potentials*” by Thölke & De Fabritiis, ICLR 2022). We chose TorchMD-NET as a second architecture for our experiments in Section 5.1 for three main reasons. First, it is a transformer-based architecture, as opposed to a GNN-based one like GNS (hence covering two families of models). Second, it is built specifically for 3D molecules with the appropriate equivariance to rotations and translations. Lastly, it is recent and has been shown to perform well compared to other baselines, yielding SOTA results. As discussed in our related work section, pre-training via denoising is architecture-agnostic and we conjecture it will improve performance on other architectures than TorchMD-NET and GNS as well. In fact, recent follow-up work [1] has already shown its effectiveness for another architecture inspired by Graphormer.
> 4. **Open source model/code.** Our code for experiments on TorchMD-NET has been included in the supplementary material. The code is open source along with model checkpoints, and a link to the repository will be added upon publication.
>
> Thank you again for your feedback! We hope that we addressed your questions. Please let us know if you have any further questions.
>
> [1] *One Transformer Can Understand Both 2D & 3D Molecular Data.* Luo et al., arXiv 2022 (https://arxiv.org/abs/2210.01765)

---

> > ### Comment · Reviewer_8vAk · 2022-11-17
> > **Thank you for the response!**
> >
> > Thank you for the response, I have no more questions.

---

### Author Response · Authors · 2022-11-17
**Response to all reviewers/AC**

Thank you very much to the reviewers for their helpful feedback. We are glad that the work was appreciated and positively received by all reviewers, and we believe that the feedback has improved it further. All changes made to the paper are shown in blue in the revised manuscript. In summary:
1. We have added Appendix B.1 (to be merged with Section 3.2.1 if accepted) to clarify the differences in the underlying assumptions of denoising for generative modelling vs. force learning (reviewer tMrH).
2. We will also rearrange the results to incorporate Tables 9-11 into the main text if accepted (reviewer YJNn).
3. We have clarified requested experimental details in Appendices E and F (reviewers 8vAk, YJNn, tMrH).
4. We have add a discussion of GNS/GNS-TAT on OC20 with additional experimental results in Appendix A.2, showing that GNS-TAT matches GNS on OC20 without re-tuning TAT hyperparameters, and we improved the main text’s figure labels for clarity (reviewer YJNn).

Please let us know if you have any further questions.

---

### Decision · Program_Chairs · 2023-01-20

**Decision:**

Accept: notable-top-25%

**Justification For Why Not Higher Score:**

The paper still has some weaknesses, including the insufficient comparison with more baselines, and the lack of experimental details (including hyperparameters) in the main body of the paper.

**Justification For Why Not Lower Score:**

The strength of the paper includes: simplicity and elegancy of the proposed method, the physical insights of denoising, and the solid experimental results.

**Metareview: Summary, Strengths And Weaknesses:**

The paper proposes a simple and efficient pretraining strategy, which predicts the added noise into 3D positions. The paper further shows the connection between denoising and molecular force field learning. Experiments demonstrate the effectiveness of the proposed method.

All reviewers speak highly of the paper. They like the simplicity and elegancy of the proposed method, the physical insights of denoising, and the solid experimental results.  Although they also mentioned some weaknesses, including the insufficient comparison with more baselines, and the lack of experimental details (including hyperparameters) in the main body of the paper. Overall speaking, the paper is clearly above the bar, and should be accepted.


**Note From Pc:**

if the above contains the word "oral" or "spotlight" please see: "oral" presentation means -> notable-top-5% and "spotlight" means -> notable-top-25%. As stated in our emails, we are disassociating presentation type from AC recommendations